# Giant electrostriction-like response from defective non-ferroelectric epitaxial BaTiO$_3$ integrated on Si (100)

Shubham Kumar Parate [1,7] ✉, Sandeep Vura [1,7] ✉, Subhajit Pal[1,2], Upanya Khandelwal [1], Rama Satya Sandilya Ventrapragada[1], Rajeev Kumar Rai [1,3], Sri Harsha Molleti[1], Vishnu Kumar[1], Girish Patil[1], Mudit Jain[1], Ambresh Mallya[1], Majid Ahmadi [4], Bart Kooi [4,5], Sushobhan Avasthi[1], Rajeev Ranjan [6], Srinivasan Raghavan[1], Saurabh Chandorkar[1] & Pavan Nukala [1] ✉

Lead-free, silicon compatible materials showing large electromechanical responses comparable to, or better than conventional relaxor ferroelectrics, are desirable for various nanoelectromechanical devices and applications. Defect-engineered electrostriction has recently been gaining popularity to obtain enhanced electromechanical responses at sub 100 Hz frequencies. Here, we report record values of electrostrictive strain coefficients ($M_{31}$) at frequencies as large as 5 kHz ($1.04 \times 10^{-14}$ m$^2$/V$^2$ at 1 kHz, and $3.87 \times 10^{-15}$ m$^2$/V$^2$ at 5 kHz) using A-site and oxygen-deficient barium titanate thin-films, epitaxially integrated onto Si. The effect is robust and retained upon cycling upto 6 million times. Our perovskite films are non-ferroelectric, exhibit a different symmetry compared to stoichiometric BaTiO$_3$ and are characterized by twin boundaries and nano polar-like regions. We show that the dielectric relaxation arising from the defect-induced features correlates well with the observed giant electrostriction-like response. These films show large coefficient of thermal expansion ($2.36 \times 10^{-5}$/K), which along with the giant $M_{31}$ implies a considerable increase in the lattice anharmonicity induced by the defects. Our work provides a crucial step forward towards formulating guidelines to engineer large electromechanical responses even at higher frequencies in lead-free thin films.

The quest for lead-free piezoelectric materials with a large electromechanical (EM) response is important for nano electromechanical systems (NEMS) devices such as actuators, ultrasonic transducers, sensors, energy harvesters, nanopositioners, and micro-robotics[1-4].

A common strategy to increase piezoelectric response from non-centrosymmetric materials is to flatten the thermodynamic energy profile easing polarization rotation[5-8]. Such an energy landscape is seen at the morphotropic phase boundary (MPB) in materials such as

[1]Center for Nano Science and Engineering, Indian Institute of Science, Bengaluru 560012, India. [2]School of Engineering and Materials Science, Queen Mary University of London, London E1 4NS, United Kingdom. [3]Materials Science and Engineering, University of Pennsylvania, 3231 Walnut Street, Philadelphia, PA 19104, USA. [4]Zernike Institute for Advanced Materials, University of Groningen, Groningen 9747AG, The Netherlands. [5]CogniGron center, University of Groningen, Groningen 9747 AG, The Netherlands. [6]Materials Engineering, Indian Institute of Science, Bengaluru 560012, India. [7]These authors contributed equally: Shubham Kumar Parate, Sandeep Vura. ✉e-mail: shubhamkp@iisc.ac.in; sandeepv@iisc.ac.in; pnukala@iisc.ac.in

Pb(Zr,Ti)O$_3$ (PZT) and relaxor ferroelectrics, and in materials with many coexisting local orders (K$_{0.5}$Na$_{0.5}$NbO$_3$-xBaTiO$_3$ (KNN-BT, for e.g.)[5–10]. In classic ferroelectrics, domain wall motion is a major contributor to the EM response, especially at lower frequencies[11,12]. In centrosymmetric systems, especially thin films, field-induced piezo-electricity has been engineered via various strategies such as creating asymmetric contacts[13], multiple interfaces[14–16] and defects[15–18].

Recently, defect-based strategies are gaining tremendous traction to engineer materials with large non-classical EM responses, especially at low frequencies (<100 Hz). "Giant" electrostriction[4] has been first reported[19] in 20% Gd-doped CeO$_2$ in 2012. It has by now been established that the response of electroactive defect complexes in non-dilute concentrations to an external electric field, and their corresponding elastic dipoles results in substantial electrostrain[14,17]. Such an effect, which is second order in nature or electrostrictive, has also been engineered in other defective oxide systems such as Nb and Y stabilized Bi$_2$O$_3$[20] La$_2$Mo$_2$O$_6$[21] and so on[4]. Electromechanical effects (both first and second order) were also observed in thin-film systems such as BaTiO$_3$ through oxygen vacancy-induced chemical expansion[18]. Colossal piezoelectric coefficients in centrosymmetric Gd-doped CeO$_2$ ($d_{33}$ ~ 200,000 pm/V) at ~1 mHz have been attributed to field-induced defect/ion motion at those frequencies[17]. In the same system, at slightly larger frequencies (100–1000 Hz) polaron hopping is shown to be responsible for an effective $d_{33}$ ~ 100 pm/V, and which is comparable to response from the classic piezoelectrics such as PZT. Furthermore, enhanced field-dependent d$_{33}$ values of ~1100 pm/V were also reported in A-site deficient NaNbO$_3$ films deposited on SrTiO$_3$ replete with out-of-phase boundaries (2D defects)[15,16,22]. Defect-driven EM response dramatically reduces as frequency increases and becomes insignificant beyond 1 kHz. A unifying thread in all these studies is to engineer EM response through large concentration of electroactive defects (0D and 2D defects), which elastically interact with each other to give a coherent and large strain response.

Here, we report record second-order EM coefficients at frequencies larger than 1 kHz ($M_{31}$ = 1.04 ×10$^{-14}$ m$^2$/V$^2$ at 1 kHz, and 3.87 ×10$^{-15}$ m$^2$/V$^2$ at 5 kHz) in heavily A-site and oxygen-deficient non-ferroelectric barium titanate (Ba$_{0.87}$TiO$_{3-\delta}$) thin film system. The BTO was epitaxially integrated onto Si with TiN as a buffer layer. The oxygen scavenging property of TiN buffer layer, along with suitable choice of growth parameters, allows us to obtain a non-stoichiometric, yet stable, perovskite phase. We show that the observed giant EM response is correlated to the defect-based mechanisms giving rise to dielectric relaxation, and possibly even to electroactive twin boundary mobility. We propose that in addition to having large defect concentration, systems with significantly enhanced defect-induced dielectric constant and relaxation behavior, are good materials for large and non-classical EM response. This work is a significant step in achieving lead-free CMOS compatible materials with giant EM response.

## Results
### Structure and defect characterization
The effect of deposition conditions on epitaxy and growth of the defective complex oxides on Si studied here was previously developed by Vura et al. and has been reported elsewhere[23,24]. A buffer layer of epitaxial TiN ((100) oriented, 40–60 nm) was deposited on n++ Si (100) using reactive pulsed laser deposition in N$_2$ (99.9999% pure) atmosphere in eclipsed off-axis configuration[23] [see Methods]. Epitaxial growth of barium titanate (BTO, <001> oriented, 175–245 nm) is enabled directly on this platform (Supplementary Fig. 1a), using standard pulsed laser deposition. All the results reported here are on samples grown at conditions described in the methods, and post-annealed in atmosphere at 500 °C. High Angle Annular Dark Field Imaging in Scanning Transmission Electron Microscopy (HAADF-STEM) and corresponding energy dispersive spectroscopy (EDS)

mapping (Supplementary Fig. 1b, c) reveals the presence of a TiO$_x$ (18–20 nm) layer between TiN and BTO, formed via interfacial redox reaction during annealing. This reaction also helps in rendering the BTO layer oxygen deficient. Oxygen vacancies can be identified as missing oxygen contrast in iDPC STEM images and corresponding line profiles (Fig. 1a). The out-of-plane and in-plane lattice parameters, obtained from $\theta$−$2\theta$ and in-plane XRD scans (Fig. 1b) correspond to 4.038 (± 0.005) Å and 4.023 (±0.005) Å, respectively. The unit cell volume of the film (65.35 Å$^3$) is larger than that of bulk ceramic BTO ($a$ = 3.99 Å, $c$ = 4.04 Å, 64.31 Å$^3$), as a result of chemical expansion due to defects[24]. EDS quantification further reveals a uniform Ba/Ti ratio of 0.86 ± 0.04 (precision error) throughout the film. Depth resolved XPS was further performed to substantiate the EDS data and understand the oxidation states of cations in our defective films. The XPS spectra obtained at 30 nm depth from the surface is shown in Fig. 1d–f. Peak fitting procedures reveal the presence of Ti in both 4+ and in reduced 3+ oxidation states, with Ba/Ti ratio of 0.87 (±0.04), very similar to the EDS quantification. Thus, the composition of our films is Ba$_{0.87}$TiO$_{3-\delta}$ (also see Supplementary Fig. 1d–f for surface XPS spectra). The polarization maps obtained from HAADF STEM (Fig. 1c) images show the presence of nano polar-like regions, similar to relaxors[25]. c/a ratios vary in the film between 0.96 and 1.08, the $c/a$ ratio maps also show presence of NPRs correlated to the regions defined in the polarization mapping (Supplementary Fig. 2a, b, and corresponding Supplementary Note 2). The non-existence of a unique polarization axis reveals that our defective BTO has a different symmetry than P4mm, replete with local disorder. Our films are also replete with 2D defects such as twin boundaries (Supplementary Fig. 2c).

### Electromechanical characterization
On these defective BTO thin films ($t$ = 170 nm), we fabricated inter-digitated electrodes (IDE, Supplementary Fig. 2d (inset), Supplementary Fig. 2e), with electrode separation of 20 μm, and finger length of 80 μm. A voltage applied across two terminals majorly contributes to in-plane electric field in BTO (referred to as *direction 1*), owing to the presence of a low dielectric constant insulating TiO$_x$ interfacial layer. In this configuration, by applying a large signal AC voltage, we measured the out-of-plane displacement (*direction 3*) on various devices (~10 devices) using laser doppler vibrometer (see Methods, also see Supplementary Fig. 2d). Voltage was cycled between $V_{max}$ (1–5 V) and $-V_{max}$ at different frequencies (1 kHz to 50 kHz), for >5000 cycles on each device (see Methods, Supplementary Note 3 on acquisition, Supplementary Fig. 3), and time-averaged strain response as a function of voltage is computed [Methods]. The piezoelectric tensor component we extract from these experiments, hence, is $d^{*}_{13}$ and the electrostrictive tensor component is $M_{31}$ (in Voigt notation). Lateral electric field ($E_1$) is estimated as $V_{applied}$/20 μm and is taken to be spatially uniform across the BTO layer in the device (both laterally and vertically). For assumptions and justification on calculation of $E_1$ (used for $M_{31}$ calculation) refer to Supplementary Fig. 4 and Supplementary Note 4.

The EM strain ($\epsilon_i$) as a function of electric field ($E_j$) can be expanded upto second order as follows:

$$\epsilon_i = d^{*}_{ij}E_j + M_{ij}E_j^2 \tag{1}$$

$$\epsilon_i = (d^{*}_{ij})_0 e^{i\varnothing_1}E_j + (M_{ij})_0 e^{i\varnothing_2}E_j^2 \tag{2}$$

where

$$E_j(\omega) = Re((E_j)_0 e^{i\omega t}) \tag{3}$$

$(d^{*}_{ij})_0$ is the amplitude of piezoelectric coefficient and $(M_{ij})_0$ is the amplitude of electrostrictive coefficient of defective BTO, $\varnothing_1$ and $\varnothing_2$

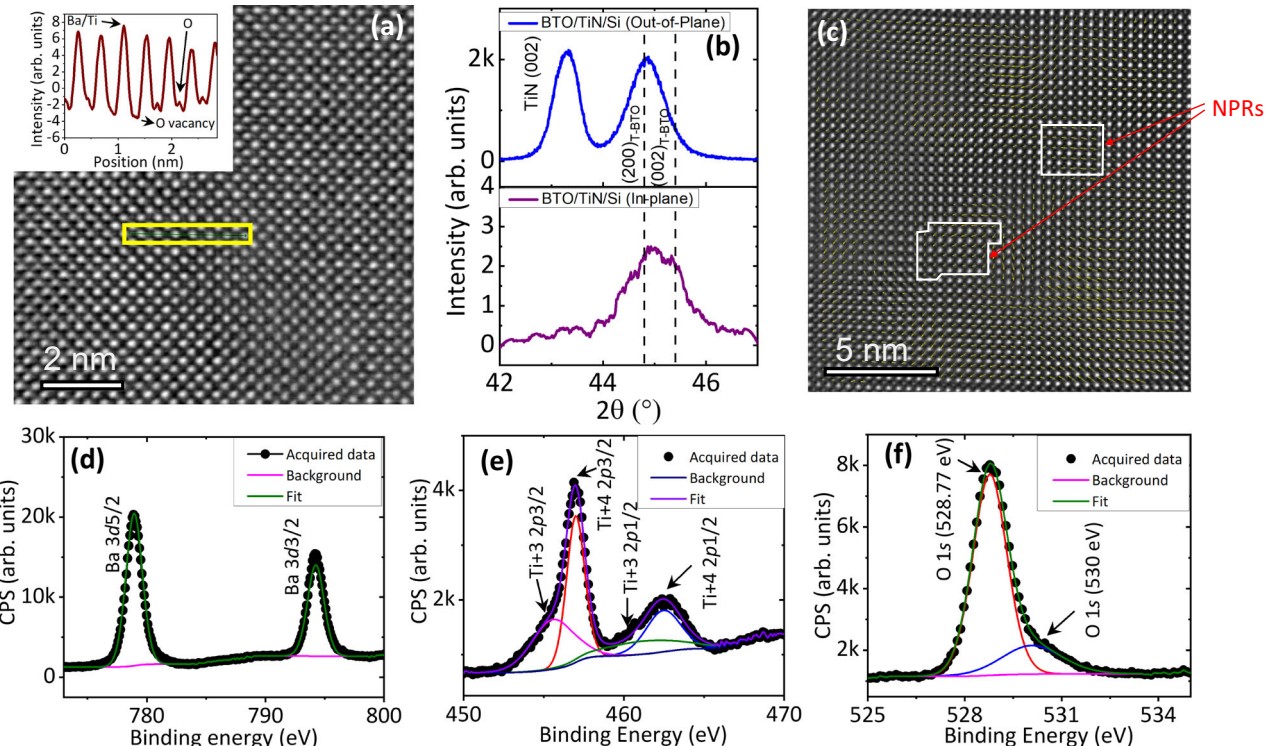

**Fig. 1 | Structural and chemical characterization of defective BTO thin films. a** iDPC STEM image and corresponding line intensity profile (inset) from the highlighted region. Yellow rectangle represents the atomic column used for line profile in inset. Low intense oxygen peaks show up in some columns and are absent in some revealing the presence of oxygen vacancies. **b** X-ray diffraction $\theta$–$2\theta$ scans showing the out-of-plane Bragg peaks (top) and in-plane scans (bottom). **c** High resolution HAADF STEM image of BTO overlayed with polarization map nano polar-like regions (NPRs) are enclosed in white boxes. High resolution XPS spectra from bulk with fits of (**d**) Ba $3d$ (**e**) Ti $2p$ and (**f**) O $1s$. Source data are provided as a source data file.

represent the phase difference between voltage ($V$) and piezoelectric strain ($\epsilon_i^{(1)}$); and $V^2$ and second order response ($\epsilon_i^{(2)}$), respectively.

The first order (harmonic) strain response is given as follows:

$$\epsilon_i^{(1)} = (d_{ij}^*)_0 e^{i(\omega t + \varnothing_1)} E_j \tag{4}$$

The second order strain response is given as follows:

$$\epsilon_i^{(2)} = (M_{ij})_0 e^{i\varnothing_2}(E_j)_0^2 (\cos(\omega t))^2 = (M_{ij})_0 e^{i\varnothing_2}(E_j)_0^2((\cos(2\omega t)+1)/2) \tag{5}$$

which can be further split into second harmonic component and a DC component as follows

$$\epsilon_i^{(2)}(2\omega) = \frac{Re\left(\left(M_{ij}\right)_0 \left(E_j\right)_0^2 e^{i(2\omega t + \varnothing_2)}\right)}{2} \tag{6}$$

$$\epsilon_i^{(2)}(DC) = \frac{Re\left(\left(M_{ij}\right)_0 \left(E_j\right)_0^2 e^{i\varnothing_2}\right)}{2} \tag{7}$$

Averaged input voltage and corresponding averaged displacement response as a function of time on lateral devices are shown in Fig. 2a (details of measurement noise and sensitivity in Supplementary Fig. 5a and Supplementary Note 5). The schematic representation of the device is also displayed in the inset of Fig. 2a. The Fourier transform of displacement-time response for a device with input AC voltage at 1 kHz is shown in Supplementary Fig. 5b. Here we see a weak first order and zero third order response, and a predominant 2nd order and a slightly weaker 4th order response. Our weak first order effects were not repeatable over days of measurements (Supplementary Fig. 5) and can also result from non-mechanical effects such as optical

interference[26]. Given that 3rd order response is also absent, we conclude that our material is not piezoelectric or very weakly piezo-electric, and thus in the rest of the discussion we do not analyze piezoelectricity in any further detail.

Figure 2b shows displacement vs time response of a device tested at 5 kHz input voltage, bandpass filtered in 4.5–11 kHz frequency range, and the corresponding FFT. This is also reflected in the butterfly-like strain-voltage plots in Fig. 2c. It is important to note that in our strain vs field plots, any residual strain at zero field is a consequence of the phase difference between strain and corresponding order of the field.

Fourier filtered second order strain response (for the device data shown in Fig. 2b) as a function of voltage is shown in Fig. 3a. The amplitude of effective electrostrictive coefficient ($|M_3|$) on a representative device, calculated from Fig. 3a as a function of frequency at 3 V and 5 V is shown in Fig. 3b. We estimate that $|M_{31}|$ at 1 kHz and $V_{max} = 5$ V is $1.04 \times 10^{-14}$ m²/V², and it reduces by more than half at 5 kHz to $3.87 \times 10^{-15}$ m²/V² (see phase as a function of frequency in Supplementary Fig. 5c). These are record values observed at frequencies >1 kHz, as can be seen from the comparison of M coefficients as a function of frequency of various giant electrostrictors in Fig. 3c[19,27–33]. | Tan $\delta|_{EM}$, the electromechanical loss tangent, as function of frequency (Fig. 3d) shows a peak at 6–7 kHz for various devices (see Supplementary Movie 1 to see how losses create butterfly-like hysteresis in electrostriction). To better compare with conventional piezoelectric materials also, we report the values of $d_{13}$-effective ($d_{13\text{-}eff}^*$) estimated as Max Strain/Max field as a function of frequency in Supplementary Fig. 5d. $d_{13\text{-}eff}^*$ is 2.57 nm/V at 1 kHz and at higher frequency such as 9 kHz, it decreases to 390 pm/V. These values are larger or comparable with Pb-based materials that show larger electrostrain[34]. It must be noted that although we report $d_{13\text{-}eff}^*$, this is to only compare our devices with conventional piezoelectric materials. We again reiterate

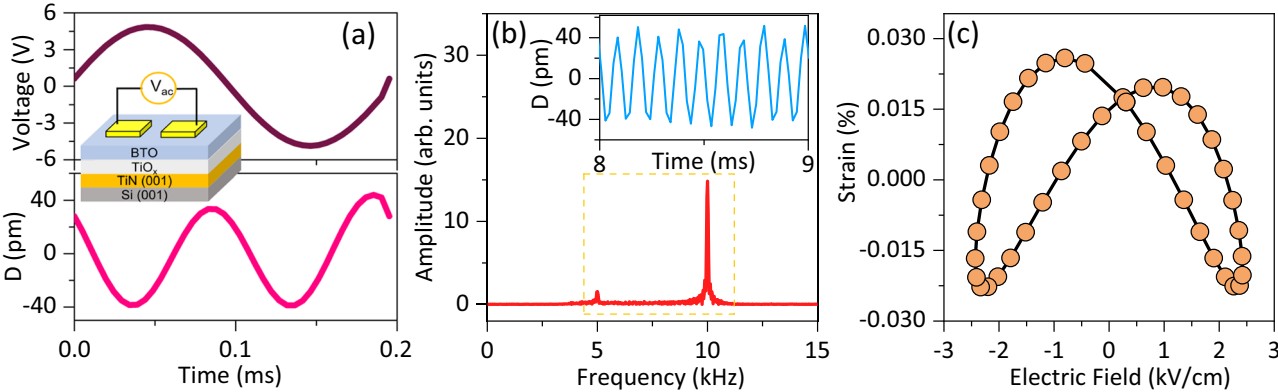

**Fig. 2 | Electromechanical response of defective BTO lateral devices. a** Averaged voltage (top) and corresponding averaged displacement (*D*) response (bottom) as a function of time and lateral measurement setup schematic (inset). Optical micrograph of IDE and corresponding dimensions (in μm) are shown in Supplementary Fig. 2(d (inset) and e). **b** Fast Fourier Transform of band pass filtered (range shown in yellow box) displacement-time response shown in inset. **c** Averaged strain-electric field response obtained from data shown in the inset of **b**. Source data are provided as a source data file.

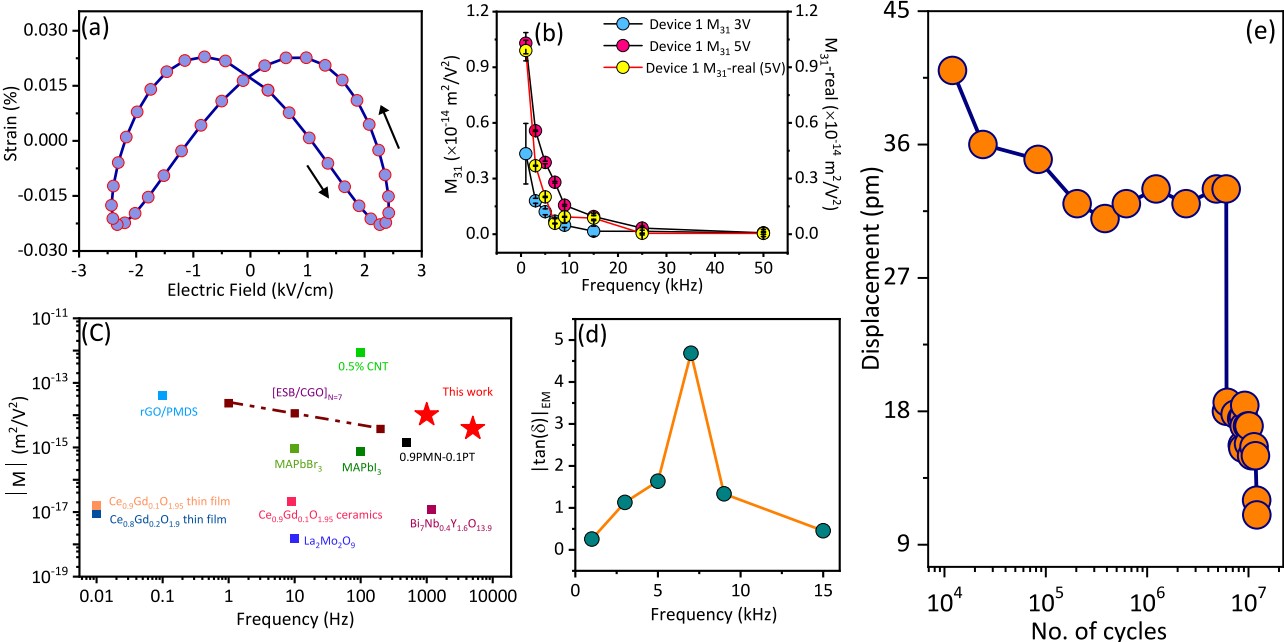

**Fig. 3 | Analyzing the 2nd order EM response. a** Fourier filtered second harmonic strain response at 5 kHz as a function of varying electric field, (black arrows represent the direction of displacement waveform with respect to voltage waveform). **b** $M_{31}$ (also $M_{31\text{-}real}$, shown in yellow) coefficients varying with frequency at two different $V_{max} = 3$ V (pink) and $V_{max} = 5$ V (sky blue), error bars represent ± standard deviation. **c** Compilation of giant *M* electrostrictive coefficients previously reported on different material systems, and comparison of the values (indicated as red stars) we achieve in this work, as a function of frequency. **d** Electromechanical loss tangent ($|\tan \delta|_{EM}$) at various frequencies, peaking at 6–7 kHz for all the tested devices. **e** Device endurance test conducted for $10^7$ voltage cycles at 5 V peak voltage and 3 kHz. Source data are provided as a source data file.

that the electromechanical effect in our devices is second order, and there is negligible piezoelectricity (or perhaps even zero).

We also note that our second order strain amplitude linearly increases with $(V_{max})^2$ upto $V_{max} = 5$ V, and saturates at larger voltages (Supplementary Fig. 5e, also see Supplementary Note 8. More importantly, our devices show signs of fatigue only after >$10^6$ cycles of operation (Fig. 3e), which is an extremely good endurance metric for MEMS based applications.

The real part of $M_{31}$, $M_{31\text{-}real}$, is the conservative, dissipation-less part of the electrostrictive coefficient, which is also a giant response (Fig. 3b). Thermodynamically $M_{ij\text{-}real}$ can also be estimated through a converse effect as the ratio of change in susceptibility ($\Delta\chi_i$) with applied stress ($X_j$). For this, we performed some preliminary

nanoindentation and bending experiments (see Supplementary Figs. 6–8). In the indentation, we apply static stress ($X_{33}$) through a nanoindenter on a lateral device ($X_3$), and measure the change in its susceptibility ($\chi_1$) (see Supplementary Fig. 6, Supplementary Note 9 for more details). This gives estimates of $M_{13\text{-}real}$ (real part) which is also in the order of $10^{-15}$ V²/m² for frequencies <10 kHz. In the bending experiments (Supplementary Figs. 7 and 8), we apply homogenous $X_1$ by bending the substrate, and then measure the change in capacitance of the vertical devices, and calculate the change in susceptibility ($\chi_3$) of the active BTO layer. This gives an estimate for $M_{31\text{-}real}$, albeit certain assumptions described in Supplementary Note 9. The indirect effect still gives a giant $M_{31}$ of ~$10^{-16}$ m²/V², which is however, an order of magnitude less than the estimates from direct electrostrain

measurements. This could be a result of clamping, and assumptions of ideal capacitors and non-consideration of leakage effects in the indirect effect estimates (see Supplementary Note 10). These results further substantiate that the off-diagonal $M$ ($M_{13}$ and $M_{31}$) tensor elements are indeed giant, and our defective BTO films can be classified as giant (off-diagonal) "$M$" electrostrictors[4].

To answer the question whether our films are also giant $Q$ electrostrictors, we performed synchronized measurements of dielectric displacement ($D_1$)- field ($E_1$) along with strain ($\varepsilon_{33}$)-field ($E_1$). Details of the measurements, data and analyses can be found in Supplementary information (Supplementary Fig. 9, Supplementary Note 11). We find that $|Q_{31}|$ measured at 5 kHz and $V_{max} = 3$ V for less-leaky devices in the order of $10^{-7}$ (m$^2$/C)$^2$. Despite measuring $D_1$ at fields and devices where leakage is small, it is possible that we still overestimate the values of $D_1$. So, our values of $|Q_{31}|$ should be interpreted as the lower limit of the polarization electrostrictive coefficients. By the definition presented by Yu and Janolin[4], our films are not giant $Q$ electrostrictors[4]. This could be related to the easily polarizable and soft matrix, which enhances $M$ but not $Q$[4]. Thus, in this manuscript, we do not further discuss values of $Q_{31}$, but only analyze the origins of $M_{31}$.

For completeness, let us also note that the electromechanical response (strain vs field) measured on vertical metal-insulator-metal capacitors is very weak, owing to the large field drops across the low dielectric constant TiO$_x$ layer (Supplementary Fig. 10).

## Electrical characterization

To understand the correlation of large EM response to dielectric and leakage properties we performed large signal AC $I$-$V$ measurements from 1 to 50 kHz with voltage varying from $-V_{max}$ to $V_{max}$ (for $V_{max} = 1$, 3 and 5 V), and small signal capacitance-voltage-frequency measurements from 1 Hz to 100 kHz. General $I$-$V$ response of what is referred to as "less leaky" device is shown in Fig. 4a (also see Supplementary Fig. 11). Below $V_{max} = 3$ V, a good dielectric behavior (capacitive response, phase difference between $V$ and $I$ is close to 90°) is observed, and leakage (resistor) characteristics begin beyond 3 V. C-$f$ data (Fig. 4b) shows a relaxation behavior, with the device capacitance reducing significantly with frequency beyond 1 kHz. |Tan $\delta$|$_D$ dielectric loss tangent shows a major peak at ~100 kHz, and a smaller hump at 6−7 kHz frequency, suggesting at least two different RC time constants in the device (Supplementary Fig. 11e). It is important to note that second order EM loss peak correlates with the dielectric loss peak at 6−7 kHz frequency. Reduction in $|M_{31}|$ from 1 to 50 kHz also correlates with the decrease in capacitance values. Furthermore C-V data also shows nonlinearities until 10 kHz, and not beyond that (refer to Supplementary Fig 11f).

To glean better insights into what dielectric relaxations are correlated to EM response in our complex system, we modeled our impedance spectroscopy data with equivalent circuit (shown in inset of Fig. 4c). Our IDE was simplified as lateral device with two terminals, with one terminal ground and the other one sourced (at voltage $V$). In this configuration, voltages at various nodes are represented in Supplementary Fig. 12a. Our system is modeled as two parallel R||C circuits, referred to as A1 and A2 (see Supplementary Fig. 12a) overall in parallel with interfacial Maxwell-Wagner[35] capacitance. A1 represents lateral BTO layer which exhibits various defect induced relaxation mechanisms (from polar nano-like regions shown in Fig. 1c, from interfacial charges at the twin boundaries[12,36] shown in Supplementary Fig. 2c, and electroactive twin boundary motion). A2 contains two different R||C elements in series. The first lumped element represents vertical field drop across BTO (voltage drop from source electrode to BTO-TiO$_x$ interface: $V-V_2$, as well as from BTO-TiO$_x$ interface to the ground: $V_2-V_3$ in Supplementary Fig. 12a), while the second element represents voltage drop vertically across TiO$_x$ layer (also see Supplementary Note 4). At frequencies <50 kHz, we obtain good fits based on such a model as shown in Fig. 4c and Supplementary Fig. 11e. Our model suggests that the dielectric relaxations in the BTO layer show a RC time constant of 6−7 kHz, at which frequency we also observed a peak in the |Tan $\delta$|$_{EM}$ response of the electromechanical behavior. Thus, the features responsible for electromechanical and dielectric response in BTO correlate well.

## Discussion

The second order EM behavior is an effect of one or more of the following phenomena: a) intrinsic lattice electrostriction, b) ferroelectric switching and other field-induced phase transition, c) thermal expansion because of device heating, d) non-classical defect-induced electrostriction. Our defective BTO does not show any ferroelectric switching up until the maximum voltages of measurement (see AC $I$-$V$ plots in Fig. 4a and Supplementary Fig. 11a−c). We also measure a large coefficient of thermal expansion (CTE), $\alpha_{33} = 2.36 \times 10^{-5}$/K, through in situ XRD measurements on these films (results published elsewhere[24], also reproduced in Supplementary Fig. 12c). Coincidentally, large CTE and large electrostriction are both related to lattice anharmonicity, in this case induced by defects. The large CTE also means that an increase in device temperature by ~15−30 K can already result in ~100 pm expansion or 0.02% strain, observed on our representative devices (Fig. 3a). Device temperatures increase owing to leakage-induced Joule heating or dielectric loss, both of which are included in the in-phase component of current with voltage in the large signal AC measurements. To understand the effect of heating, we simulated the device temperature rise using electrothermal modeling via LTspice. The effective electrothermal circuit is shown and described in Supplementary Fig. 12b. The devices were modeled as linear resistors, with resistance ($R$) given by the ratio of maximum voltage to maximum current. In reality, our devices are non-linear resistors, and this underestimates the resistance, and eventually overestimates the

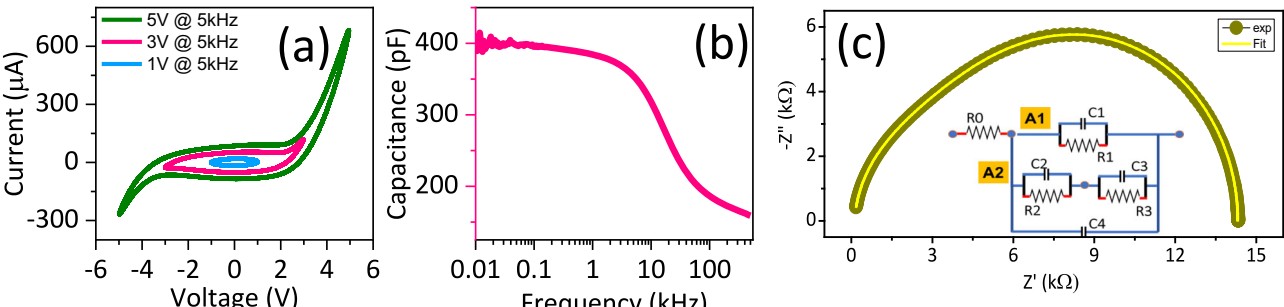

**Fig. 4 | Large signal AC I-V characterization and impedance measurements. a** Large signal I-V characteristic on inter digitated electrodes (IDE) at three different voltages ($V_{max} = 1, 3, 5$ V). Leakage characteristics begin beyond $V_{max} = 3$ V. **b** Small signal capacitance measurement as a function of frequency. **c** Measured impedance and corresponding fit of the equivalent circuit (inset), detailed circuit is shown in Supplementary Fig. 12a). Source data are provided as a source data file.

device temperature range. The $V^2/R$ power was fed into the thermal circuit as an input heat source, when voltage was cycled at various frequencies from $-V_{max}$ (3 V and 5 V) to $V_{max}$. We show that for a representative "less leaky" device whose *I-V* characteristics are shown in Fig. 4a (also see Supplementary Fig. 11a–c for *I-t* at various $V_{max}$), the max device temperature rise is 0.8–0.9 K at 1 kHz and 5 V (see Supplementary Fig. 13a), and it reduced with increase in frequency (see Supplementary Fig. 13b).

Next, we tested the electromechanical response of devices which are leakier than the representative device (leaky *I-V* and their corresponding $M_{31}$ vs frequency shown in Supplementary Fig. 14a–d). For the leaky device presented in Supplementary Fig. 14b ($t = 170$ nm, EM response in Supplementary Fig. 14d), the max displacement measured at 5 V and 1 kHz is 500 pm. Our simulations show that, at these conditions the device temperature amplitude is at the maximum 10 K (Supplementary Fig. 15a), which contributes to only about 60 pm of the displacement out of 500 pm measured (also see Supplementary Note 14, Supplementary Figs. 16 and 17 for effect of substrate heating in the measured strain oscillations). Supplementary Fig. 14e shows consolidated data of effective $M_{31}$'s measured at 5 kHz on several devices of different leakage current densities, fabricated on films of thicknesses 60 nm, 120 nm, 170 nm and 240 nm. We do not find any overall trends of $M_{31}$s with leakage current densities and film thicknesses. These results clearly show that major contributor to large EM response is electrostriction, and not device heating or ferroelectric switching.

When 5 V is applied across the IDE, the field ($E_1$) in the BTO layer can be approximated to 2.5 kV/cm (Supplementary Fig. 4c, d). At such a small field, we obtain 100 s of pm vertical displacement, resulting in giant $M_{31}$s. We note that the decrease in $|M_{31}|$ with frequency is correlated with the decrease in device capacitance in the 1–50 kHz frequency range. Furthermore, dielectric and electromechanical loss tangents also correlate peaking at 6–7 kHz. Our impedance spectroscopy modeling reveals that such a behavior is a consequence of relaxation mechanisms occurring in polar nano-like regions induced relaxations and at the twin boundaries of the BTO layer. Furthermore, the mobility of electroactive twin walls also contributes to the enhanced electromechanical responses as is the case in ferroelectric materials[12]. It may be noted that the fundamental origin of all these relaxations and the correlated giant electrostriction-like behavior at these frequencies is due to structural and polarization disorder created by Ba and oxygen non-stoichiometry, and associated lattice anharmonicity (also evidenced by large CTE). The giant $|M_{31}|$ coefficients at larger frequencies are a consequence of large electroactive defect induced polarizabilities, coupled elastic dipoles and long-range coherent strain fields in addition to possible electroactive twin wall motion.

In this work, we demonstrate giant electrostrictive coefficients upto 5 kHz for defective barium titanate films epitaxially integrated with Si. These are record values reported on any materials system beyond frequencies of a few 100 s of Hz. We also show that such large response is robust, and that fatigue does not set in even upto a few million cycles. We show that these coefficients and corresponding EM losses are very much correlated with dielectric relaxations induced by various defects, which fundamentally introduce large lattice anharmonicity. This lets us propose that in order to achieve giant EM responses at even higher frequencies, it is worth to first explore defect-engineering strategies aimed towards reducing the RC time constants of defect-induced dielectric relaxation mechanisms. In addition, we also propose that in addition to point defects, 2D defects that are mobile (just as in ferroelectrics) help in achieving larger electrostrictive responses at higher frequencies. Our work provides a significant step forward in expanding the bandwidth of giant EM responses in Si compatible, lead-free materials.

## Methods

### Synthesis of epitaxial stack of defective BaTiO$_3$/TiN on Si

**Stage 1.** An epitaxial TiN template on Si is deposited first. 40–60 nm epitaxial TiN on n++ Si (100) was deposited by reactive pulsed laser deposition (ablating a Ti target (99.5% pure, GfE, GmBH, Germany) in $N_2$ (99.9999% pure) ambient in an eclipsed off-axis configuration[23]. The Si substrate is cleaned using acetone, iso-propyl alcohol and HF/$H_2O$ dip (10%v/V) for 2 mins. The cleaned Si sample is loaded into the deposition chamber and evacuated to a pressure of $5 \times 10^{-6}$ mbar. The Si sample is heated resistively to the desired temperature, passing direct current trough it i.e., the substrate is heater. The heater ramp rate is 200 °C/min in $N_2$ ambient at 0.6 mbar, to reduce the formation of amorphous $SiO_2$ which enables the epitaxy of TiN on Si (100)[23]. The substrate temperature, target to substrate distance and the chamber pressure during TiN deposition are 700 °C, 3.0 cm and 0.6 mbar, respectively. The laser fluence and repetition rate are 1.5 J/cm$^2$ and 20 Hz, respectively.

**Stage 2.** The 175–245 nm BaTiO$_3$ was deposited on the epitaxial TiN/Si(100) by PLD in a different chamber equipped with reflection high energy electron diffraction (RHEED) (STAIB Instruments, GmBH, Germany, Model: Torr RHEED) operated at 30 kV to monitor the growth surface. Prior to deposition the TiN/Si is dipped in HF:$H_2O$ (1:10 V/V) for 30 s to remove the TiO$_x$ formed on the surface[23]. The laser fluence and repetition rate are 1.5 J/cm$^2$ and 2 Hz, respectively. The substrate temperature and chamber pressure are 600 °C and $5 \times 10^{-6}$ mbar, respectively during the deposition and cooled to room temperature in an oxygen ambient at a pressure of 0.1 mbar. Post deposition the sample is annealed at 500 °C for 1 h at atmospheric pressure with an oxygen flow rate of 3 slm.

### Structural and composition characterization

**X-ray diffraction and X-ray photo-emission spectroscopy.** The crystal structure of BTO/TiN/Si was investigated using 4-circle X-ray diffractometer using a Cu-K$_\alpha$ source (1.5402 Å) (Rigaku Smart Lab). Chemical composition of the BTO films was investigated using XPS (Kratos Axis Ultra) equipped with a monochromatic Al X-ray source. To determine the BTO film stoichiometry, a 10 mm circular disc of bulk BTO pellet was used as a reference during the XPS. The survey and high resolution XPS spectra were acquired with 1 eV and 0.1 eV resolution, respectively. The in-situ XPS depth profiling for BTO/TiN/Si samples and BTO pellet was done using an Ar$^+$ ion beam with an energy of 4 kV. C1s peak (284.6 eV) was used to calibrate the survey and high resolution XPS spectra. BTO film composition was calculated from survey spectra and the background data was modeled using Shirley algorithm.

**STEM imaging and analysis.** The cross-section FIB lamella for TEM analysis was prepared using a focused ion beam (Model: FEI, Scios2) and is investigated using a double aberration corrected Thermofisher Themis microscope operated at 300 kV, and a non-aberration corrected Themis microscope also operated at 300 kV, equipped with chemi-STEM EDS system. STEM-EDS was performed on non-aberration corrected Themis microscope, and data was acquired until sufficient counts (SNR > 5) was obtained from binned pixels. IDPC-STEM images were obtained using a four-segment anuular bright field detector collecting signal from 6 to 20 mrad. This is a linear imaging technique with contrast ~Z, and thus is sensitive to lighter elemental columns such as oxygen.

**Polarization mapping analysis and oxygen vacancy identification.** Atomic resolution HAADF-STEM images were used to map Ti displacements in every unit cell. To estimate and quantify such displacements a clean scan distortion free lattice image was selected, which were then Bragg filtered using Digital Micrograph for final

mapping. Finally, Ti displacement mapping away from the center of mass of Ba (A-site) unit cell was performed using Atomap and Temul-toolkit. Finally, these displacements are represented by arrows (indicating both magnitude and direction of displacement) overlaid on the image. Ti displacements are a good estimator for unit cell dipole moments in $BaTiO_3$.

**Device fabrication.** The patterns for electrical contacts were defined using optical lithography (Heidelberg) and Cr/Au (5/60 nm) contacts were deposited using DC sputtering. For out of plane measurements the contact pad size of $100 \times 100 \, \mu m^2$ was used. For in-plane measurements interdigitated contacts which consists of 4 pairs of fingers with length of $80 \, \mu m$, width of $20 \, \mu m$ and spacing between the fingers of $20 \, \mu m$ were used (Supplementary Fig. 2d (inset) and e).

**Electromechanical response and I-V characteristics.** Laser Doppler Vibrometer (LDV) (model: MSA 500) equipped with a 532 nm reference and probe laser was used for studying both in plane and out of plane electromechanical response of the $BaTiO_3$ films. Displacement response for different frequencies was obtained for 30–40 msec as a function of time is averaged over several cycles (250–500). This response at each frequency was further averaged to obtain single waveform with standard deviations shown as error bars in M coefficient plots. So all in all, every tested device was cycled for >5000 cycles depending on the frequency of operation to obtain one data set as shown in Fig. 2.

Frequency dependent *I-V* and Impedance measurements were performed using MFIA Impedance Analyzer. Nyquist plots were fitted using z-fit utility of EC-Lab Demo. To estimate the rise in device temperature, electro-thermal simulations were carried out using LTspice simulator (Analog Devices).

## Data availability
Source data are available in figshare repository.

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

## Acknowledgements

This work was partly carried out at Micro and Nano Characterization Facility (MNCF), and National Nanofabrication Center (NNfC) located at CeNSE, IISc Bengaluru, funded by NPMAS-DRDO and MCIT, MeitY, Government of India; and benefitted from all the help and support from the staff. P.N. acknowledges Start-up grant from IISc, Infosys Young Researcher award, and DST-starting research grant SRG/2021/000285. The authors acknowledge funding support from the Ministry of Human Resource Development (MHRD) through NIEIN project, from Ministry of Electronics and Information Technology (MeitY) and Department of Science and Technology (DST) through NNetRA and the Thematic Unit of Excellence for Nano Science and Technology project from DST Nano Mission. The authors thank Prof. Praveen Kumar from IISc for allowing us to conduct the thermal imaging experiments using his lab equipment, Dr. Vijeyandra Shastry for all the help in performing them, and Venu Bhat for help with the indentation experiments. PN would like to acknowledge discussions with Evgenios Stylianidis from University College London. The authors also acknowledge Rishabh Navaneet for the supplementary movie 1. All the authors acknowledge the usage of national nanofabrication center, micro nano characterization center, and advanced facility for microscopy and microanalysis of IISc for various fabrication and characterization studies.

## Author contributions

Ideation began with discussions between PN, SV, SP. SKP, PN, SV, SP and UK designed the experiments. SV synthesized the samples. SKP, SP, SV, UK, SHM carried out the electromechanical measurements. SKP, RKR, MA, AM, SV carried out STEM analysis, with data analysis routines run by RKR and SKP. SV carried out XRD, XPS and corresponding data analysis. UK, SKP, RSS carried out impedance spectroscopy, impedance device modeling and electrothermal modeling using SPICE with supervision from SC and PN. GP, MJ performed complimentary COMSOL electro-thermal simulations and independently confirmed the SPICE results. VK, SV fabricated the devices. BK, PN, MA supervised STEM analysis; SA, SC, PN, RR supervised the electrical and electromechanical measurements and corresponding analysis; SR, PN, RR supervised the growth and characterization aspects. PN managed the coordination between the various teams. SKP, PN, SV, SP cowrote the manuscript with inputs from UK and RSS. All the authors read and commented on the manuscript.

## Competing interests

The authors declare no competing interests.
