## [Peer Review File · Nature Communications]

Giant electrostriction-like response from defective non-ferroelectric epitaxial BaTiO₃ integrated on Si (100)REVIEWER COMMENTS

Reviewer #1 (Remarks to the Author):

The manuscript is very interesting and potentially worthy of publication in NC. However, the claim of M31 reaching $>10\text{-}15 \text{ m}^2/\text{V}^2$ will be met with a lot of skepticism, especially in view of the facts:

1. the films have interfacial layer and the field applied is not clear: the text says the that thickness of the films is $<245 \text{ nm}$ and the minimum voltage applied 3 V , which places the voltage to $> 12 \text{ MV/m}$. This is three orders of magnitude above the values stated on Figure 2c and Fig. 3c. Figure S3c shows fields of $\pm 300 \text{ kV/cm}$, which is 30 MV/m adding to the overall confusion.
2. To exclude thermal expansion and avoid misinterpreting it as an electrostriction effect, the simulation is definitely not enough, due to a number of reasons, the simplest of which is that the films may have large variation in lateral conductivity. The only way to prove it is to use thermal imaging, which require some special equipment and skills.
3. M31 depends on the field applied and it INCREASES with the field, while it should be expected to do exactly opposite if the saturation is approached. The increase suggests complicated interface effects, which may render the thermal modelling to be unreliable.

In view of the above and in order to promote a potentially very interesting paper, I strongly advise measuring converse the electrostriction effect (deform the substrate and measure the change in the dielectric permittivity. The elastic modulus of BTO is well known and does not change significantly by the presence of defects. Since, this method does not require application of external voltage (apart for a few mV to measure the dielectric constant), it will be far more reliable. Moreover, since measuring the dielectric permittivity it is possible to cover the whole frequency range. Superimposing the direct and converse effects on Fig. 3b(must have log Y in the revised version!) will prove that the rest of the data are correct.

The numerical estimate is: the strain from Figures S3c x elastic modulus of BTO x electrostriction coefficient reported / $\epsilon_0 > 100$. Such a change in dielectric permittivity is impossible to miss.

If the converse effect supports the data all other deficiencies are not important and the manuscript can be recommended for publication full heartedly. However, without these data, it cannot be published as it will stir more confusion than good.

Reviewer #2 (Remarks to the Author):

What are the noteworthy results?

Vura et al. report on a lead-free oxide thin films, Si-compatible, that present remarkable electrostrictive performances at high frequencies. The presented results, once consolidated, represent a significant contribution to the promising field of electrostriction as a potential complement or replacement of piezoelectricity. Non-conventional electrostrictors have emerged over the last ten years but have been plagued so far by drastic frequency limitations, preventing their use for ultrasonic applications; the results presented in this article contribute to overcome this hurdle. The durability of the response, though, is only demonstrated over a few thousands cycles, far below application standards.

Will the work be of significance to the field and related fields? How does it compare to the established literature? If the work is not original, please provide relevant references.

The work is of significance along .. axes:

- It provides a new type of defect-induced electrostrictive material that exhibits superior performance at higher frequencies than currently available materials.

- It demonstrates the ability to integrate such materials on Si, bringing the electrostrictive electromechanical systems closer to their integration in the semiconductor industry.

Does the work support the conclusions and claims, or is additional evidence needed?

The reported values of the electrostrictive coefficients require calculating a deformation and an electric field from a displacement and an applied voltage. In this regard, several questions arise:

1. How has the deformation been calculated? It remains unclear what are the hypotheses underlying this calculation, in particular, the mechanical and electrical boundary conditions.

2. How was the electric field calculated? Interdigitated electrodes on thin films tend to lead to complex field lines. In that regard, how has the maximum field value been chosen, e.g. in Fig.2 and 3? If the entire voltage difference applied between the interdigitated electrodes generates a field with parallel field lines, the resulting field is one to two orders of magnitude larger than the one reported. In addition, what is the dependence on the thickness of the films, and how can it be explained

In addition, I have questions about the electrostrictive coefficients themselves:

1. The dependence of the M electrostrictive coefficient on the applied voltage suggests a non-linearity of the field dependence of the induced dielectric displacement. Has such a measurement been made? In addition, if M may depend on the electric field amplitude, the Q tensor, relating strain to the (square of the) field-induced polarisation component of the dielectric displacement, does not exhibit such

dependence. Providing a figure equivalent to Fig.3b but for Q31 would ascertain the electrostrictive nature of the response.

2. The authors report a larger M coefficient in leaky samples. This seems counter-intuitive as part of the electrostatic energy will be converted in a current rather than in an electric field. Such lower electric field should lead to a lower displacement but not necessarily decrease the electrostrictive coefficient. The more significant defect concentration mentioned in lines 240-242 is a possible explanation, but it equates defects and conduction, which is not necessarily obvious to me and should present a dependence upon the thickness of the film.

3. The induced deformations correspond to a compression (negative strain). This is the case for most non-conventional electrostrictors. Have the authors any explanation for the sign of the corresponding electrostrictive coefficient?

4. The authors considered the effect of thermal expansion and were right to do so, as a quadratic response does not necessarily imply that the electrostrictive response is the main response mechanism. Have the authors measured the induced strain as a function of increasing voltage amplitude? If so, is the maximum induced strain proportional to the square of the electric field?

5. The FFT of the induced strain exhibits a main peak at 10kHz, which is twice the frequency of the exciting voltage, which is consistent with electrostriction. There is a minor contribution at 5kHz. What is the proposed origin of this piezoelectric component? Have the authors carried out a Fourier analysis of the voltage applied to the sample (rather than the one generated by the waveform generator)? In addition, how do the authors explain the absence of a Fourier component at 20kHz, 40kHz, and so on? Electromechanical responses tend to exhibit higher-order components (odd ones for piezoelectricity and even ones for electrostriction). I was expecting the large electrostrictive response reported here to follow such a trend.

Are there any flaws in the data analysis, interpretation, and conclusions? Do these prohibit the publication or require revision?

Is the methodology sound? Does the work meet the expected standards in your field?

Rather than actual flaws, some aspects of the data analysis, interpretation and conclusion lead to the following remarks:

1. The claim about the robustness of the response seems a bit hasty. A few thousand repetitions is far from a fatigue measurement, requiring at least a hundred thousand cycles.

2. In addition, using such material for sensing applications (as mentioned in the introduction) requires the ability of the thin films to withstand the application of a bias field and the determination of an effective g or d piezoelectric coefficient. Considering the insulating properties of the thin films, this claim would need to be substantiated by specific experiments.

3. The strain at zero field is not zero, neither in Fig.2 nor in Fig.3. What causes this unexpected reference point?

4. What is the noise level of the strain measurements? Fig.S3(c) is extracted from the displacement reported in Fig.S3(b) but as the FFT weight spectrum is not provided, the quadratic curve is not certain to be related to electrostriction. Such a noise level measurement could be carried out with no voltage applied and considered in the data analysis of such low-level signal.

Such remarks only require revision.

Is there enough detail provided in the methods for the work to be reproduced?

The description of how the displacement measurements have been made could be clearer. As far as I understand it, the measurement time was kept more or less constant (30 to 40 msec) and repeated over 250-500 cycles for averaging purposes. The following sentence (starting on line 329) states that “this response at each frequency was further averaged to obtain single waveform”. I do not understand this sentence. Would the authors care to reformulate?

The choice to make measurements over finite time rather than over a given number of periods changes the time step used in the Fourier transform and therefore affects the precision of the Fourier transform.

The manuscript is unclear on the samples on which the measurements have been carried out. Several samples with varying thicknesses have been investigated and the manuscript would gain clarity by clearly stating whether the presented measurements are an average over various samples, and whether the structural, electromechanical, dielectric and thermal investigations were carried out on the same sample, on a collection of samples, etc.

Once these comments have been addressed, this manuscript presents interesting, novel results on a topic of great interest and therefore should be published in Nature Communications.

Reviewer #3 (Remarks to the Author):

The authors report a study on defect engineered barium titanate finding a marked increase in electrostrictive strain coefficients at frequencies as high as 5kHz. The authors show that the observed giant electromechanical response is correlated to defect-based mechanisms giving rise to dielectric relaxation. Importantly, these films are epitaxially grown on Si and exhibit robust behavior when cycled

greater than 5000 times, which can substantially impact devices requiring lead-free, large electromechanical responses. These findings are supported by structural characterization (STEM, XRD), XPS, laser doppler vibrometry, and electrical measurements.

The manuscript is well written, and findings are well supported. Furthermore, the findings presented in the current manuscript are relevant and important to the broader ferroelectrics community; however, there are a few issues that prevent me from recommending it for publication in the current form. Below are the detailed comments.

1. In Figure 1, the authors perform STEM, XRD, and XPS to extract the local polarization, lattice parameters, and composition, respectively. It would be useful to also extract the local c/a ratio via STEM analysis to further understand how local lattice variations are correlated with oxygen vacancies and nanopolar regions.
2. XPS only probes several nanometers at most. This coupled with the fact the films were annealed in air post growth raises concerns that the surface composition differs significantly from the bulk. Please provide discussion/analysis/additional data supporting the claims of composition.
3. The authors state that the defective BTO reported in this manuscript is non-ferroelectric. The basis for this claim is the absence of switching in the AC I-V curves. However, Figure S5b shows some “leaky” ferroelectric like switching characteristics. Furthermore, the authors only probe $\pm 6V$; is it possible the coercive voltage is above the $\pm 6V$ probing bias? Including additional characterization to verify the non-ferroelectric claims is vital (e.g. Polarization-electric field measurements/PUND, piezoresponse force microscopy, etc.).

Minor comments:

1. Line 65: keep Gd:CeO₂ labels consistent (line 65 vs 58).
2. Figure 3: some axes labels overlap tick labels.

We profoundly thank all the reviewers for their positive feedback, insightful comments, and specific experimental suggestions. We are very impressed by the quality of the peer review, which is exemplified by the high-quality comments, and genuine intention of all the experts to see a much-improved manuscript. From our end, we implemented a lot of these suggestions, and we are glad to report that we are able to substantiate and further strengthen our claims of giant and robust electrostriction (in terms of M coefficients, cyclable to more than million times) on Si integrated defective BaTiO₃ thin films.

Reviewer #1 (Remarks to the Author):

The manuscript is very interesting and potentially worthy of publication in NC. However, the claim of M_{31} reaching $>10\text{-}15 \text{ m}^2/\text{V}^2$ will be met with a lot of scepticism, especially in view of the facts:

1. the films have interfacial layer and the field applied is not clear: the text says the that thickness of the films is $<245 \text{ nm}$ and the minimum voltage applied 3 V , which places the voltage to $> 12 \text{ MV/m}$. This is three orders of magnitude above the values stated on Figure 2c and Fig. 3c. Figure S3c shows fields of $\pm 300 \text{ kV/cm}$, which is 30 MV/m adding to the overall confusion

Response: In Fig 2c and 3 of the manuscript, we report the values of in-plane electric field (E_x or E_1) applied across BaTiO₃, by using an IDE geometry, with a separation between the source electrode and ground of $20 \mu\text{m}$. The values reported are estimated as $V_{\text{applied}}/20 \mu\text{m}$, which assumes a uniform E_1 across the device (justification provided at the end of this response). When applied voltage amplitude is 5V , the field E_1 was estimated as 2.5 kV/cm . In Fig 2c, we represented these values accurately, however, in Fig 3a, there was a typo in the units. kV/cm was mistyped as kV/m , and we corrected this mistake.

Given the existence of an interfacial layer (TiO_x) and a floating electrode (TiN), a natural question arises, whether the mechanical displacement of the film we are measuring is due to E_3 (field in the vertical direction) or E_1 (in-plane component)? In the IDE lateral geometry, given that the source is at a potential V , ground at 0 , TiN can be assumed to be floating at $V/2$. As a result, the total vertical field evolves spatially along the horizontal (l or x) direction, from $E_{3, \text{max}}$ to $-E_{3, \text{max}}$, passing through 0 at the center. The actual COMOSL simulation result of the E_3 field profile as a function of horizontal spatial coordinate (l or x) is presented in Fig R1a. Any contribution from E_3 to the mechanical displacement should hence be inhomogenous, and we checked that our device response is very homogenous when measured closer to the source, at the center and closer to the ground as shown in Fig R1b. Note that the laser spot size (probe volume) is $\sim 4\text{-}5 \mu\text{m}$.

Fig R1 (a) vertical field E_3 or E_y field variation laterally (b) displacement response from lateral field corresponding to the points (marked as 1, 2 and 3) shown in IDE schematic is homogenous .

To convince ourselves further that our response in IDEs is a result of in-plane field E_1 and not the inhomogenous out-of-plane field (E_3), we directly performed experiments on vertical MIM capacitors with Pt and TiN as the top and bottom electrodes, and BaTiO₃ and the TiO_x layer being in series to each other. These results were shown in Fig S3b (Now revised to Fig S8b and c), but the explanation was scant in the earlier version of the m/s inviting confusion. We apologize for that. As the referee pointed out, in this geometry it is possible for the fields to go to 100 s of kV/cm. The results in Fig S8c, however, suggest that when 5 V is applied across the top and bottom electrode at 5 kHz, the membrane displacement is very small (amplitude ~ 15 pm). In our IDE configuration, E_3 (at 5V between source and ground) is much less than in the vertical devices, and so will not contribute to any significant displacement.

An associated question is if the value of M_{33} can be estimated from the experiments described in Fig S8c. For this, we performed impedance spectroscopy across vertical devices (Fig R2a), modelled them with an equivalent circuit as shown in Fig R2b, with a low impedance BTO layer in series with a high impedance TiO_x layer. Based on our models and fits (performed over a few devices), we see that for an applied voltage of 5 V at 5 kHz across the top and bottom electrode, the voltage drop across BTO layer (170 nm thick) is ~ 0.8 V, which corresponds to, a vertical field amplitude $E_{30}=47$ kV/cm in this device. The second order strain component due to this field is given as follows:

$$Strain_{DC} + Strain_{AC} = \frac{M_{33}}{2} E_{30}^2 (1 - \cos(2\omega t))$$

Fig S8a, b and c (also shown here as Fig R2c, d and e), shows AC strain variation at 10 kHz (2nd order) for voltage input at 5 kHz. We note that the amplitude of oscillation is ~ 15 pm. This corresponds to M_{33} of $1.02 \times 10^{-19} \text{ m}^2/\text{V}^2$. All the devices we measured had M_{33} in the same order

of magnitude, which is close to its classical value. Thus, the giant electrostriction we report here is only in terms of M_{31} and for lateral devices, not for M_{33} . Now, we added this discussion in the **supplementary note 11 (Fig S8)**. [1][2].

Fig R2 (a) Small signal impedance on MIM device and nyquist fit using an equivalent circuit in (b); (b) shows the voltage drop across BTO/ TiO_x layer calculated at 5kHz, (c) Metal insulator metal (MIM) capacitor schematic (d) Vertical displacement as a function of time of MIM capacitor with BTO thickness 170 nm and (e) second harmonic response with a varying electric field (5 V) at 5 kHz and small signal impedance fit on IDE device geometry and voltage drop in (f) and (g) respectively.

Finally, we justify our assumption that E_1 (in plane component) is uniform spatially from source to the ground in the BTO layer. For this, we first fit our impedance spectroscopy data, performed on lateral IDEs by modelling the system using equivalent circuit as shown in Fig R2f, g (same in Fig 4c and Fig S10a-and described in manuscript). Using the extracted impedance values of different layers as the input, we performed COMSOL simulations to evaluate the in-plane field (E_1) profile spatially in the BTO layer across the two electrodes, along a certain 2D cross-sections of the IDE Fig R3a). Here we report results on this geometry (Fig R3b), with BTO layer thickness of 170 nm, TiO_x layer thickness of 15 nm, and interelectrode lateral spacing of 20 μm. We see that larger the dielectric constant of BTO layer, more uniform is the spatial distribution of E_1 (or E_x). We take the ϵ_r of TiO_x as 15, which is consistent both with literature as well as our own impedance spectra of vertical devices. BTO layers ϵ_r is about 100 times larger, however, leakage (or a parallel resistive path) makes it a layer with much lower apparent impedance. This renders the effective $\epsilon_{r\text{-apparent}}$ of BTO layer (estimated from $|Z| = 1/\omega \times C_{\text{apparent}}$ as 1.79×10^8) to be 3×10^4 times greater than that of non-leaky BTO. Our simulations show that for BTO films of 170 nm thickness, such a large $\epsilon_{r\text{-apparent}}$ renders a spatially uniform E_1 from source to ground, whose value can simply be evaluated as $V_{\text{appl}}/20 \text{ um}$. (Fig R3c). E_1 also does not vary much vertically (out-of-plane) except near the TiO_x interface (Fig R3d).

Please also refer to response given to question2 of referee 2.

Fig R3 (a) IDE schematic showing the cross section taken for COMSOL simulation (b) the corresponding cross section (c) horizontal field E_1 or E_x variation laterally across the IDE for various apparent dielectric constants. We note that at large dielectric constants, E_1 is spatially homogenous (d) E_1 variation with depth starting from the BTO surface for a 170 nm thick film. We see that E_1 is uniform across the thickness of BTO and starts to drop at the interfacial layer.

We now include this discussion and data in the supplementary information (**Note 4, Fig S4c and d**). We hope that our experiments are clearer now and will not cause any confusion among the readers.

2. To exclude thermal expansion and avoid misinterpreting it as an electrostriction effect, the simulation is definitely not enough, due to a number of reasons, the simplest of which is that the films may have large variation in lateral conductivity. The only way to prove it is to use thermal imaging, which require some special equipment and skills.

Response: Taking inspiration from this comment of the referee, we attempted some thermal imaging experiments using an IR camera on our IDE devices as shown in Fig R4a. AC voltage signal with amplitude of 5V was applied at 100 Hz, and DC temperature rise of the device (upto the steady state) was measured over seconds (and not the AC temperature variation that should happen at 200 Hz). We see that the DC saturation temperature rise on our non-leaky devices is < 1 K (Fig R4b).

We then utilize the fact the DC component of temperature rise (Fig R4c) upto saturation (ΔT_{DC}), and its AC amplitude (ΔT_{AC} , see Fig R4c) should be related, since the source of heating is the same current (I). We performed electrothermal LT-Spice simulations on our devices to understand relation between ΔT_{AC} and ΔT_{DC} . For a device with thermal capacitance of 6.037×10^{-7} J/K, on which an input AC current waveform of amplitude 6.87×10^{-4} A is applied at 1 kHz, temperature variation with time is shown in Fig R3c. These simulations were performed for different input current amplitudes and device thermal capacitances. In Fig R4c, we compile ΔT_{DC} as a function of ΔT_{AC} at all these simulated conditions. At relatively larger thermal capacitances (which explain our experimental data), we see that the AC amplitude of temperature oscillations is less than or comparable to DC rise in temperature (Fig R4c). Given that, ΔT_{DC} in our experiments is <1 K, we can conclude that ΔT_{AC} should be smaller than 1 K, consistent with our earlier modelling which suggested an AC temperature amplitude of <0.8 K in all our non-leaky devices. These preliminary thermal imaging experiments supported by simulations further suggest that thermal expansion is not a major factor in our results.

We are currently setting up a better measurement schemes to measure temperature rise at a faster frame rate, so that we can capture images at milli second time scales and understand the dynamic oscillations of temperature directly with AC signal. However, as the referee pointed out, we are still learning the skills, and this experiment will take much longer. Thus, we show these preliminary results only to the referees.

Fig R4: (a) IR image recorded on the latter device by applying 5 V AC at 100 Hz. (b) Temperature change with time recorded on the sample at two different points. (c) rise in temperature with time (showing ΔT_{DC} and ΔT_{AC}), at a device thermal capacitance of 6.037×10^{-7} J/K and applied current of 6.87×10^{-4} A, (d) enlarged view of saturated region in (c) showing AC variation after DC saturates (e) AC vs DC temperature rise at various values of thermal capacitances

3. M31 depends on the field applied and it INCREASES with the field, while it should be expected to do exactly opposite if the saturation is approached. The increase suggests complicated interface effects, which may render the thermal modelling to be unreliable.

In view of the above and in order to promote a potentially very interesting paper, I strongly advise measuring converse the electrostriction effect (deform the substrate and measure the change in the dielectric permittivity. The elastic modulus of BTO is well known and does not change significantly by the presence of defects. Since, this method does not require application of external voltage (apart for a few mV to measure the dielectric constant), it will be far more reliable. Moreover, since measuring the dielectric permittivity it is possible to cover the whole frequency range. Superimposing the direct and converse effects on Fig. 3b (must have log Y in the revised version!) will prove that the rest of the data are correct.

Response: This is a very interesting and inspirational suggestion. We performed static indentation at different loads on our samples using a 30 nm radius tip in an atomic force microscope. At every load, we measured capacitance vs frequency data on our device. However, these experiments were performed on lateral devices with a 800 nm nanogap between the source and ground (Fig R5a), and not our regular IDE geometry. The reasons to adopt a nanogap geometry are twofold:

- (i) With a tip radius of 30 nm, we wanted to maximize the active area of the device that is loaded. This required minimizing the device lateral size to nanometric scale. In a limited time available, using e-beam lithography the smallest devices we were able to obtain had 800 nm gap size (Fig R5b). We acknowledge that there is further room for device geometry improvement, which is why at the current moment we do not give any rigorous quantification of the material stress state, but rather approximate the stress to be applied load/tip area.

Fig R5 (a) optical image of the nanogap device geometry (b) AFM image of device separation used for nanoindentation having 800 nm gap

(ii) Having decided on a nanogap device geometry, the next question we answered is, what should be the electrode width (w) to gap length (l) ratio that would enable us to measure changes in capacitance upon application of load, given that our impedance analyser has sensitivity in pF. Here we show our simple calculations:

From thermodynamics (Maxwell's relations), M can also be defined as follows [1]:

$$M_{ijkl} = \frac{\partial(\chi_{ij}\epsilon_0)}{\partial X_{kl}} \dots\dots\dots(i)$$

$$\frac{MA}{d} = \frac{\partial(\chi \epsilon_0)}{\partial X} \cdot \frac{A}{d}$$

$$\frac{MA}{d} = \frac{\partial(C)}{\partial X}$$

Where χ is the susceptibility, ϵ_0 is the absolute permittivity and X is the stress applied. Hence, as the referee correctly pointed out, the aim of this experiment is to measure changes in susceptibility (capacitance), with the application of stress (static).

$$\frac{MA}{d} = \frac{\partial(\chi \epsilon_0)}{\partial X} \cdot \frac{A}{d}$$

$$\frac{MA}{d} = \frac{\partial(C)}{\partial X}$$

$$\Delta C = \frac{MA \Delta X}{d}$$

We see from our preliminary indentation measurements that a load of 7 μN , which approximately corresponds to 2.4 GPa of stress, elastically deforms our BTO films, and the film regains its shape once the load is removed. We did not want to increase the load any further given the possibility of plastic deformation or even tearing the film. Given the maximum $\Delta X=2.4$ GPa (Youngs modulus=80 GPa [3]), we constrain our A/d such that ΔC is measurable by our system (pico Farads). For M in the order of $10^{-15} \text{ m}^2/\text{V}^2$, A/d should be $\sim 0.4 \mu\text{m}$.

In a nanogap geometry with $d = 800 \text{ nm}$ (fixed by constraints mentioned in point (i)), based on lithography and lift off constraints, A was set to be $5 \mu\text{m} \times 200 \text{ nm}$.

However, in a lateral geometry, we can apply stress in the vertical direction through nanoindenter (X_{33}) and measure lateral capacitance (χ_{11}). From this we were able to estimate M_{1133} (M_{13} in Voigt notation). Applying in-plane stress to measure M_{31} is tricky. Although, we are designing experiments to do this, we will not be able to show this data currently (in such a short time). In the following, we will show that our starting experiments already show that M_{13} is also giant. We believe given the underlying lattice anharmonicity and dielectrically soft matrix (large tensor elements in susceptibility and mechanical compliance) it should not be surprising to expect that the off-diagonal tensor elements M_{13} and M_{31} are correlated.

We used AFM with cantilever tips of radius 30 nm for nano indenting BTO at static loads of 0.1 (35 MPa) and 7 μN (2.4 GPa), between the lateral electrodes. At every load we measured capacitance vs frequency (Fig R6a) using an impedance analyser (MFIA-Zurich instruments).

For a load of 7 μN (stress applied is 2.4 GPa) the estimated $\Delta C = 3 \text{ pF}$, for $M=10^{-15} \text{ m}^2/\text{V}^2$, and for 0.1 μN , ΔC is practically 0. Our C vs f plots between 9 and 11 kHz (Fig R6b) show that ΔC measured is indeed 2-3 pF at 7 μN load. At higher frequency (65-80 kHz), we do not see any change in the capacitance upon loading (Fig R6c), which also consistent with our interferometry (LDV) measurements of M coefficient at higher frequencies (less than the system sensitivity, see also Fig R13, and corresponding response to referee 2 about noise and instrumental sensitivity).

Also worth noting is the fact that we performed these experiments also on reference samples (SiO_x nanogap devices) and did not see any changes in capacitance with load of 7 μN (Fig R6d). As we can see from Fig R6d that the noise in capacitance measurement is about 0.5 pF, which puts a limit on the sensitivity of M from the indentation experiments to be $\sim 2 \times 10^{-16} \text{ m}^2/\text{V}^2$ with the current geometry. Any M less than this will not be measurable, as we see for data on our devices at higher frequencies (60-80 kHz), and also on the reference samples.

Fig R6 (a) shows capacitance variation with frequency at two different loads, to view the subtle changes enlarged (b) view (low frequency) and enlarged (c) view (high frequency) from plot (a), (d) Nanoindentation on SiO_x using similar nanogap devices showing no change in capacitance with loading.

While these set of experiments already show another proof of BTO being a giant M electrostrictor, still only a part of the film sees the load of the indenter. There is further room for improvement before quantifying the M_{13} values that we obtain from these experiments, by a) using larger radius indenters with flat tips, and (b) further scaling down the device channel size by optimizing e-beam lithography. We are currently pursuing these measurements and will aim for standalone work from this. At this stage, however, we include these preliminary measurements which support our conclusions in the supplementary information (**Note 9**, **Fig S6**).

The numerical estimate is: the strain from Figures S3c x elastic modulus of BTO x electrostriction coefficient reported / $\epsilon_{00} > 100$. Such a change in dielectric permittivity is impossible to miss.

If the converse effect supports the data all other deficiencies are not important, and the manuscript can be recommended for publication full heartedly. However, without these data, it cannot be published as it will stir more confusion than good.

Response: We believe that our nanoindentation experiments in the current stage do support our conclusions. We sincerely thank the current referee for suggesting us these measurements, and in some way also setting a new direction to our research. We will continue this effort and come up with a more rigorous and quantitative analysis of both M_{13} and M_{31} coefficients as our upcoming work.

References

- [1] Jiacheng Yu, Pierre-Eymeric Janolin; Defining “giant” electrostriction. *Journal of Applied Physics* 7 May 2022; 131 (17): 170701
- [2] Hideki Ogihara, Thesis, Structure - Property – Compositional Relationship in the BaTiO₃ – BiScO₃ Ceramic System, 2008
- [3] Kohei Maruyama *et al* 2022 *Jpn. J. Appl. Phys.* **61** SN1011

Reviewer #2 (Remarks to the Author):

What are the noteworthy results?

Vura et al. report on a lead-free oxide thin films, Si-compatible, that present remarkable electrostrictive performances at high frequencies. The presented results, once consolidated, represent a significant contribution to the promising field of electrostriction as a potential complement or replacement of piezoelectricity.

Response: We thank the referee for recognising the importance of these results and for raising pertinent issues that we have addressed now.

Non-conventional electrostrictors have emerged over the last ten years but have been plagued so far by drastic frequency limitations, preventing their use for ultrasonic applications; the results presented in this article contribute to overcome this hurdle. The durability of the response, though, is only demonstrated over a few thousand cycles, far below application standards.

Response: We performed fatigue measurements now on the devices, which was missing earlier and show that the devices can endure to greater than 10^6 cycles before they fatigue (Fig 3e in the revised manuscript). We show the data and discuss this in a later question raised by the current referee.

Will the work be of significance to the field and related fields? How does it compare to the established literature? If the work is not original, please provide relevant references.

The work is of significance along .. axes:

- It provides a new type of defect-induced electrostrictive material that exhibits superior performance at higher frequencies than currently available materials.
- It demonstrates the ability to integrate such materials on Si, bringing the electrostrictive electromechanical systems closer to their integration in the semiconductor industry.

Does the work support the conclusions and claims, or is additional evidence needed?

The reported values of the electrostrictive coefficients require calculating a deformation and an electric field from a displacement and an applied voltage. In this regard, several questions arise:

1. How has the deformation been calculated? It remains unclear what are the hypotheses underlying this calculation, in particular, the mechanical and electrical boundary conditions.

Response: We use laser doppler vibrometer (used as a double beam laser interferometer) to measure the out of plane displacement (δ_3) of BTO film. Strain, (X_{33}) is estimated as δ_3/t where t is the thickness of BTO. Mechanically BTO film is clamped to the substrate with a chemically formed ~ 15 nm TiO_x interlayer and a 60 nm TiN layer separating it from the substrate. Electrically we use an IDE lateral electrode (Fig R7a) geometry by sourcing one electrode (at voltage V), and grounding another. There is a bottom electrode also, TiN, which is floating at $V/2$. However, this bottom electrode is separated from BTO with a low dielectric constant TiO_x layer ($\epsilon_r=15$) (schematic Fig R1b).

Fig R7 (a) IDE geometry for applying lateral field, (b) schematic showing the stacking along with IDE electrodes with source and ground.

2. How was the electric field calculated? Interdigitated electrodes on thin films tend to lead to complex field lines. In that regard, how has the maximum field value been chosen, e.g. in Fig.2 and 3? If the entire voltage difference applied between the interdigitated electrodes generates a field with parallel field lines, the resulting field is one to two orders of magnitude larger than the one reported. In addition, what is the dependence on the thickness of the films, and how can it be explained

Response: Electric field in the horizontal direction (E_x or E_1) is simply calculated as

$$E_1 = \frac{\text{Voltage (V)}}{\text{Separation of IDE electrodes (d)}}$$

In Fig. 2 and Fig. 3, the units of field are kV/cm, with the field corresponding to maximum applied voltage (5V) being 2.5 kV/cm. In Fig 3a, however, we have mistakenly labelled the units as kV/m, which we corrected now. That E_1 is uniform is an assumption, we justify with COMSOL simulations now (next paragraph). However, given the existence of a bottom electrode, an E_3 field profile also develops. As explained for question 1 of referee 1, we rule out through other experiments that E_3 contributes to any significant BTO membrane displacement.

Here, we justify our assumption that E_1 (in plane component) is uniform spatially from source to the ground in the BTO layer. For this, we first fit our impedance spectroscopy data, performed on lateral IDEs by modelling the system using equivalent circuit as shown in Fig R8a and b (also shown in Fig 4c, Fig S10a and Fig S9e and described in manuscript). Using the extracted impedance values of different layers as the input, we performed COMSOL simulations to evaluate the in-plane field (E_1) profile spatially in the BTO layer across the two electrodes, along a certain

2D cross-sections of the IDE (mentioned in Fig R8c). Here we report results on this geometry (Fig R8d), with BTO layer thickness of 60 nm, 120 nm, 170 nm and 240 nm, TiO_x layer thickness of 15 nm, and interelectrode lateral spacing of 20 μm. We take the ε_r of TiO_x as 15, which is consistent both with literature as well as our own impedance spectra of vertical devices. BTO layers ε_r is about 100 times larger, however, leakage (or a parallel resistive path) makes it a layer with much lower apparent impedance. This renders the effective ε_{r-apparent} of BTO layer (estimated from $|Z| = 1/\omega C_{apparent}$ as 1.79×10^8) to be 3×10^4 times greater than that of non-leaky BTO. At this large apparent dielectric constant of the BTO film, the field (E₁) becomes uniform across the device (V_{applied}=5 V), for all BTO thicknesses, making our assumption of estimating E₁ as 5 V/20 μm (2.5 kV/cm) very reasonable. E₁ also does not vary much vertically (out-of-plane) except near the TiO_x interface (Fig R8e, also in SI now as Fig S4d).

Fig R8 (a) Small signal impedance fit on IDE device geometry and (b) voltage drop (c)The IDE schematic showing where the cross section (in red dotted line) has been taken for COMSOL

simulations (d) lateral field (E_1 or E_x) variation across two lateral electrode and how increase in dielectric constant (as effective $\epsilon_{r\text{-apparent}}$ is high) makes the field distribution uniform (e) distribution of field lines E_x/E_1 variation vertically at different film thickness.

In addition, I have questions about the electrostrictive coefficients themselves:

1. The dependence of the M electrostrictive coefficient on the applied voltage suggests a non-linearity of the field dependence of the induced dielectric displacement. Has such a measurement been made? In addition, if M may depend on the electric field amplitude, the Q tensor, relating strain to the (square of the) field-induced polarisation component of the dielectric displacement, does not exhibit such dependence. Providing a figure equivalent to Fig.3b but for Q31 would ascertain the electrostrictive nature of the response.

Small signal C-V data presented in Fig S9f, clearly shows a relaxor-like non-linear C-V characteristics up until 10 kHz (which also corresponds to the frequencies where M_{31} is large). This is a signature of dielectric non-linearity, as the referee correctly expected. We again present this data here as Fig R9a for the convenience of the referee.

Before discussing about Q_{31} , here we mention our attempts to obtain M through a converse measurement, as reviewer 1 suggested: i.e. applying stress (load), and measuring change in susceptibility (or capacitance) in our devices. Applying in-plane stress to measure M_{31} is tricky. Although, we are designing experiments to do this, we will not be able to show this data currently (in such a short time). As response to referee 1, question 3, we show that our starting experiments already show that M_{13} is also giant. We believe given the underlying lattice anharmonicity and dielectrically soft matrix, it should not be surprising to expect that the off-diagonal M_{13} and M_{31} are correlated. Thus, our films are giant M-electrostrictors.

We also estimated Q_{31} of our devices, as the current referee suggested, by measuring P-E loops and ϵ (strain)-E loops on the same device. Given that P is a non-linearly dependent on E, we use the following mathematical formulation to extract Q:

$$\epsilon = QP^2$$

If

$$\begin{aligned} E &= E_0 \sin(\omega t) \\ P &= P_0 \sin(\omega t) + P_1 \sin(2\omega t), \end{aligned}$$

given that the material does not have any spontaneous polarization. We address each of these terms as P_0 , $P_{2\omega}$, and so on. Second order strain $\epsilon_{2\omega}$ is contained in QP_0^2 . Therefore, in phasor notation, we have:

$$\epsilon_{33,0}(2\omega) = Q_{31} e^{-i\varphi} P_0^2$$

In Fig. R9c and d, we show the variation of second order strain (obtained from ϵ -t plots) with first order dielectric displacement (obtained from D-t), for $V_{\max}=1$ V and 3V at 5 kHz. We restrict ourselves to $V_{\max}=3$ V, because beyond that leakage becomes significant, and estimation of D will be erroneous. We obtain $|Q_{31}|$ in the order of 10^{-7} (C/m²)². By the definition of giant electrostrictors presented in reference [1] (see Fig. R9b), **our thin films are not giant Q electrostrictors**, and in

our understanding giant “M” electrostrictors need not be giant “Q” electrostrictors (see ref [1]). This could be related to the easily polarizable and soft matrix, which enhances M but not Q [1]. Thus, in this manuscript, we do not further discuss values of Q_{31} , but only look at M_{31} .

Fig R9 (a) Capacitance variation as a function of voltage at different frequencies varies non-linearly (b) definition of giant Q electrostrictors (Ref: “Defining Giant Electrostriction” [1], variation of second order strain (obtained from ϵ -t plots) with first order polarization (obtained from P-t), for (b) $V_{max}=1$ V and (c) 3V at 5 kHz

2. The authors report a larger M coefficient in leaky samples. This seems counter-intuitive as part of the electrostatic energy will be converted in a current rather than in an electric field. Such lower electric field should lead to a lower displacement but not necessarily decrease the electrostrictive coefficient. The more significant defect concentration mentioned in lines 240-242 is a possible explanation, but it equates defects and conduction, which is not necessarily obvious to me and should present a dependence upon the thickness of the film.

We now did experiments on a larger set of samples with different thicknesses, thanks to this question from the referee. However, it is not straight forward to directly compare between devices across thicknesses, without considering how leaky these devices are. In Fig. R10 (presented in Fig S12e, now), we compile all the data, in the form of current density (leakiness) vs M_{31} measured at

5 kHz and 3 V, for IDEs on films of various thicknesses (60 nm, 120 nm, 170 nm, 240 nm). Given these large statistics, we do not see any clear trends of M_{31} with leakiness. Data on devices on 170 nm thickness, seemed to show some trend with current density, and hence we claimed that leakiness increases M_{31} . Thanks to the referee's question, and our new thickness dependent measurements, we now pedal back on our claim that leakiness increases the M coefficient. The trend is not clear, with both current density and thickness of the film. All we can say at this stage is M_{31} for various samples at 3 kHz and 3V is in the order of $\sim 10^{-15} \text{ m}^2/\text{V}^2$. This provides another clue that Joule heating does not play a significant role in the observed second order electrostrain.

Furthermore, our electrothermal simulations (Fig S13) also support that most of the displacement arises from electrostriction. Our simulations show that, at these conditions the device temperature amplitude is at the maximum 10 K (Fig S13a), which contributes to only about 60 pm of the displacement out of 500 pm measured. Now we reflect this fact in the manuscript and remove any discussion on defects and leakiness. We also sincerely thank the referee for bringing up this point and suggesting us to gather more statistics on devices of various thicknesses. We now include this discussion in the manuscript (section: **Discussion**) and SI (Note 12, Fig S12e).

Fig R10 M_{31} as a function of current density of measured on various thicknesses.

3. The induced deformations correspond to a compression (negative strain). This is the case for most non-conventional electrostrictors. Have the authors any explanation for the sign of the corresponding electrostrictive coefficient?

We see both compression and tension in displacement. We report a complex electrostrictive coefficient with both amplitude and phase. By convention, we always take the amplitude $|M_{31}|$ to

be positive. If the phase lag of M_{31} with respect to E_1^2 is 180° , then in a conventional sense the electrostrictive coefficient is negative (or we see inverted parabola in strain vs E plots). If phase difference is 0, then we have a positive electrostrictive coefficient (upward parabola in strain vs E plots). Any other phase lag gives butterfly curves as we reported. So in our opinion, phase and amplitude of M_{31} represent complete information about electromechanical transduction, than the sign of M_{31} .

We now add a plot of phase lag vs frequency on a representative non-leaky IDE device (Fig R11), and we indeed see that the second order strain lags more and more with the square of the applied field with increasing frequency. The nonzero (non- π) phase difference result in hysteresis (butterfly loops), corresponding to losses in electromechanical transduction. This also points out to the role of defects (relaxing in kHz range) in such unconventional electrostriction. Now this discussion and data is shown in Fig S5c

Fig R11. Shows the phase difference between square of voltage and second order displacement as a function of frequency.

4. The authors considered the effect of thermal expansion and were right to do so, as a quadratic response does not necessarily imply that the electrostrictive response is the main response mechanism. Have the authors measured the induced strain as a function of increasing voltage amplitude? If so, is the maximum induced strain proportional to the square of the electric field?

The second order displacement vs voltage measured at various maximum AC voltages is shown in Fig R12a, and the maximum induced strain as a function of square of the applied voltage amplitude is shown in Fig R12b. As the referee correctly predicted, until 5V, we see that the induced strain goes as voltage square, and then saturates. Now we add this data and discussion in SI (see Fig S5e, Note 8).

Fig R12 (a) Displacement as a function of voltage (b) Maximum strain as a function of square of the applied voltage amplitude

5. The FFT of the induced strain exhibits a main peak at 10kHz, which is twice the frequency of the exciting voltage, which is consistent with electrostriction. There is a minor contribution at 5kHz. What is the proposed origin of this piezoelectric component? Have the authors carried out a Fourier analysis of the voltage applied to the sample (rather than the one generated by the waveform generator)? In addition, how do the authors explain the absence of a Fourier component at 20kHz, 40kHz, and so on? Electromechanical responses tend to exhibit higher-order components (odd ones for piezoelectricity and even ones for electrostriction). I was expecting the large electrostrictive response reported here to follow such a trend.

Here we present data on d_{13} measured from first order component and M_{31} on the same device at various frequencies measured on three consecutive days. The first order component is very weak (Fig R13a) and is not consistent, unlike the second order response (Fig R13b). As the referee suggested, we input our waveform through the waveform generator (5 kHz, 5 V), and measured the signal that the device experiences through an oscilloscope. And we see that in the measured voltage signal from the oscilloscope, there is only one fundamental frequency and no higher harmonics, suggesting that the 2nd harmonic displacement we measured is indeed a 2nd order response (see Fig R13c).

Fig R13 (a). The magnitude of first order piezoelectric component, d_{13} measured as a function of frequency on three different days keeps changing while in (b) the electrostrictive M_{31} response is very consistent and as a result we do not focus on the d_{13} s (c) applied voltage signal measured using external oscilloscope and corresponding (d) Fast Fourier spectrum.

We believe that the reasons for the first order component can be many fold: a) dynamic internal fields arising out of reorganization of PNR-like regions and twin boundaries leading an inconsistent piezoelectric response, and b) optical refractive index changes with applied electric field or the electrooptic effect. In our laser-based measurement, since the BTO membranes are transparent, if their refractive index changes with field (Pockels effect) these effects get reflected in the first order component of optical path difference measurement (see ref [2] for e.g. where we discuss these effects). BTO is indeed known to be a material with good Pockels coefficient [3]. As a result, we do not concentrate much on the origin of the weak first order response here.

We also profoundly thank the referee for raising the issue of “not observing” the more than 2nd order frequencies. In the earlier version, we used a band-pass filter to only pick up frequencies upto slightly above the second harmonic. We now clarified this point in the manuscript. Furthermore, we present results at 1 kHz (5 V AC signal) by adjusting the filter to capture higher order effects too, upto 10 kHz. Indeed, we see a weak first harmonic (1 kHz), strong second harmonic (2 kHz), no third harmonic (3 kHz), and a non-zero fourth harmonic (stronger than the

first harmonic) signals (Fig R14). This exactly follows the referee's prediction that odd components are weak or zero, and even components arising out of electrostriction are observed. We now include this data and discussion in the manuscript (Section: Electromechanical response) and SI (see Fig S5b).

Fig R14. Fourier analysis of the displacement signal (inset) obtained for an input voltage signal of 5V, 1kHz, showing strong 2nd and 4th harmonics, and weak first order and no 3rd harmonics.

Are there any flaws in the data analysis, interpretation, and conclusions? Do these prohibit the publication or require revision? Is the methodology sound? Does the work meet the expected standards in your field?

Rather than actual flaws, some aspects of the data analysis, interpretation and conclusion lead to the following remarks:

1. The claim about the robustness of the response seems a bit hasty. A few thousand repetitions is far from a fatigue measurement, requiring at least a hundred thousand cycles.

Here we present the data to support the robustness of our device. We have now performed device endurance measurement on our IDE devices by cycling the device to more than 10^7 cycles, by applying an AC signal of 5V amplitude at 3 kHz. We tracked the maximum displacement amplitude under tension with the cycle number (Fig R15). We see that our devices easily endure upto a few million cycles. In the representative device shown in Fig R15, fatigue sets in after 6×10^6 cycles, and the device amplitude progressively reduces. Now, we add this data to the manuscript as Fig 3e.

Fig R15. Device endurance test conducted for more than 10^7 cycles while the device can sustain its response upto 6 million cycles. The response drops sharply to half its response after 6 million cycles.

2. In addition, using such material for sensing applications (as mentioned in the introduction) requires the ability of the thin films to withstand the application of a bias field and the determination of an effective g or d piezoelectric coefficient. Considering the insulating properties of the thin films, this claim would need to be substantiated by specific experiments.

The voltage dependence of M_{31} shows a saturation beyond 6V (Fig R12, Fig S5e), and so in most of the devices we did not apply a voltage beyond 6 V. To test the maximum voltage these devices can sustain, we tested a couple of devices upto 8V, when the electrode started to peel off, possibly due to Joule heating related effects (all our devices show leakage beyond 5 V).

The effective d_{13} (d_{13}^*) coefficients were estimated as max strain/ max field. We show the variation of d_{13}^* as a function of frequency at 5V for the device whose M_{31} data is shown in Fig 3b. d_{13}^* is 2.57 nm/V at 1kHz and at higher frequency such as 9 kHz, it diminishes to 390 pm/V. These values are larger or comparable with Pb-based materials that show larger electrostrain [4]. We now include this discussion in the manuscript (Section: Electromechanical characterization), and in SI (Fig S5e).

Fig R12 Effective d_{13} as a function of frequency, value at 1kHz is 2.57nm/V which is larger than d_{13} of lead-based materials.

3. The strain at zero fields is not zero, neither in Fig.2 nor in Fig.3. What causes this unexpected reference point?

We plot a time averaged strain at every voltage, as a function of voltage. The phase difference between strain and E^2 results in non-zero strain at zero field. Phase difference manifests as hysteresis and originates from electromechanical transduction losses. We mention this point now in the manuscript (**Section: Electromechanical characterization**).

4. What is the noise level of the strain measurements? Fig.S3(c) is extracted from the displacement reported in Fig.S3(b) but as the FFT weight spectrum is not provided, the quadratic curve is not certain to be related to electrostriction. Such a noise level measurement could be carried out with no voltage applied and considered in the data analysis of such low-level signal.

Fig S8b and c (Fig R13a and b) shows displacement vs time and strain vs E_3 at 5 kHz, and $V_{\max}=5$ V. The max displacement is ~ 25 pm, which is much smaller than 80 pm measured on lateral devices. We now add the FFT in the supplementary information, from which we can see the prominent second order peak at 10 kHz which is well above the noise floor of FFT (Fig R13c, d).

Furthermore, we also measured the noise floor of the instrument at various frequencies as suggested by the referee and plot the amplitude of noise as a function of frequency (Fig R13e. Beyond 2 kHz, noise is <2 pm. Noise starts increasing below 1 kHz, which is a feature of the LDV, and this is why we do not report any measurements below 1 kHz. Our measured displacements at 1 kHz are in the order of 100s of pm across various IDE devices (3 and 5 V), which is why we were comfortable in reporting them. We now include this data and discussion in the methods and supplementary information (Fig S5, Note 5).

Fig R13 (a) Displacement from MIM vertical device at 5 kHz, 5V (b) Corresponding strain as a function of field, (c) Fast fourier transform of raw signal (in (a)) (d) Fast fourier transform of the measurement noise floor corresponding to 5kHz (no voltage applied) (e) shows the noise level as a function of frequency without applying any voltage.

Such remarks only require revision.

Is there enough detail provided in the methods for the work to be reproduced?

The description of how the displacement measurements have been made could be clearer. As far as I understand it, the measurement time was kept more or less constant (30 to 40 msec) and repeated over 250-500 cycles for averaging purposes. The following sentence (starting on line 329) states that “this response at each frequency was further averaged to obtain single waveform”. I do not understand this sentence. Would the authors care to reformulate?

We apologize for any confusion. Here, we explain the complete data acquisition and analyses scheme (see Fig R14, also Fig S3, Note 3).

Fig R14 Data acquisition and analyses explained step-by-step with an example of an input AC waveform at 9 kHz.

- (i) We apply input waveform in the form of burst chirp for 30% (9-12 ms) of the total time duration (30-40 ms) as shown in (i) in the schematic above.
- (ii) We measure the corresponding displacement waveform as shown in (ii). This displacement is averaged over 250-500 bursts.
- (iii) This burst-averaged displacement response is filtered to extract first and second harmonic responses separately.
- (iv) The burst-averaged displacement contains displacement data acquired over many voltage cycles (for e.g., at 9 kHz, and ~9 ms acquisition, voltage vs time contains ~81 voltage cycles). The first harmonic and the second harmonic displacement values are further averaged across all these cycles at discrete number of voltage points, to obtain displacement vs voltage or strain vs field plots.

We now add this procedure to the **methods section** and explain it in the SI (**Note 3, Fig S3**).

The choice to make measurements over finite time rather than over a given number of periods changes the time step used in the Fourier transform and therefore affects the precision of the Fourier transform.

To understand the errors that crop up due to finite time measurement, we quantify the “quality, Q-factor”, of the displacement oscillations defined as $\omega/\Delta\omega$, where ω defines the peak position of the forced oscillation in FFT, and $\Delta\omega$ defines the width of the peak. Q factor increases (Fig R15) with increasing frequency due to the fact that overall no. of cycles increases with frequency at fixed time ($t=40$ msec). However, even at 2 kHz, the Q factor is about 50, which gives a 40 Hz precision in measuring the oscillator frequency using FFT. This precision keeps improving with frequency.

Fig R15: Q factor dependence on frequency for more or less finite time of measurement

The manuscript is unclear on the samples on which the measurements have been carried out. Several samples with varying thicknesses have been investigated and the manuscript would gain clarity by clearly stating whether the presented measurements are an average over various samples, and whether the structural, electromechanical, dielectric and thermal investigations were carried out on the same sample, on a collection of samples, etc.

Now, we presented even more electromechanical data on films of various thicknesses (120 nm,

170 nm, 240 nm). This thickness data, as a function of current density (leakage), is compiled in Fig R10-Fig S12e, and it presents the statistics across all the measured devices. On all these devices the M_{31} coefficients measured are in the order of $10^{-15} \text{ m}^2/\text{V}^2$, at 5 kHz. We consider devices of a certain thickness and current density (at 5V) to behave similarly. The dielectric measurements presented in Fig 4, are carried out on the same device (170 nm) whose electromechanical properties were measured in Fig 2. Electrical data on two different leaky devices (170 nm) is presented in supplementary Fig S12a and b, and corresponding electromechanical behavior is also shown on the same two devices. Thermal simulations were performed using the IV data of device shown in Fig 4a of the m/s. This represents a non-leaky device. Thermal simulations on leaky devices are presented in Fig S13a and b. The input IV data for these simulations are taken from the leaky device presented in Fig S12.

Structural measurements (XRD, TEM, XPS) were done on a sample of thickness 240 nm.

Once these comments have been addressed, this manuscript presents interesting, novel results on a topic of great interest and therefore should be published in Nature Communications.

Reference

[1] Jiacheng Yu, Pierre-Eymeric Janolin; Defining “giant” electrostriction. *Journal of Applied Physics* 7 May 2022; 131 (17): 170701

[2] <https://doi.org/10.48550/arXiv.2306.14367>

[3] Abel, S., Eltes, F., Ortmann, J.E. *et al.* Large Pockels effect in micro- and nanostructured barium titanate integrated on silicon. *Nature Mater* **18**, 42–47 (2019)

[4] S. Sivaramakrishnan, P. Mardilovich *et al.* Electrode size dependence of piezoelectric response of lead zirconate titanate thin films measured by double beam laser interferometry. *Appl. Phys. Lett.* 23 September 2013; 103 (13): 132904

Reviewer #3 (Remarks to the Author):

The authors report a study on defect engineered barium titanate finding a marked increase in electrostrictive strain coefficients at frequencies as high as 5kHz. The authors show that the observed giant electromechanical response is correlated to defect-based mechanisms giving rise to dielectric relaxation. Importantly, these films are epitaxially grown on Si and exhibit robust behavior when cycled greater than 5000 times, which can substantially impact devices requiring lead-free, large electromechanical responses. These findings are supported by structural characterization (STEM, XRD), XPS, laser doppler vibrometry, and electrical measurements.

The manuscript is well written, and findings are well supported. Furthermore, the findings presented in the current manuscript are relevant and important to the broader ferroelectrics community; however, there are a few issues that prevent me from recommending it for publication in the current form. Below are the detailed comments.

1. In Figure 1, the authors perform STEM, XRD, and XPS to extract the local polarization, lattice parameters, and composition, respectively. It would be useful to also extract the local c/a ratio via STEM analysis to further understand how local lattice variations are correlated with oxygen vacancies and nanopolar regions.

Here we present the polarization mapping and c/a ratio mapping on one of the high-resolution STEM images of BTO in Fig R16a and b respectively. This is also now added to the SI as Fig S2a and b. The arrows in Fig R16a represents both the direction and the magnitude of Ti displacement and the colorbar in Fig R16b corresponds to the magnitude of c/a ratio varying from 0.96 to 1.08. Although c/a maps also indicate the existence of nanoregions and correlate spatially with the NPRs shown by polarization mapping, one-to-one correlation between Ti displacements (defining NPR) and c/a ratio is not very clear. For e.g. we do find NPRs where $c/a > 1$, yet the polarization is in-plane (R1, not tetragonal symmetry), and some where $c/a < 1$ with in-plane polarization (R3).

Fig R16 (a) Polarization map overlaid on HAADF STEM image also showing NPRs in white boxes (b) *c/a* ratio mapping on the same HAADF image with colorbar showing minimum (as blue) and maximum magnitude (as yellow) of *c/a* ratio respectively. Example NPRs are marked.

2. XPS only probes several nanometers at most. This coupled with the fact the films were annealed in air post growth raises concerns that the surface composition differs significantly from the bulk. Please provide discussion/analysis/additional data supporting the claims of composition.

This is a very valid concern. We apologize that the explanation of XPS data was scant in the earlier version. The presented XPS data was after *in situ* etching the sample for 30 sec using 4 kV Ar⁺ ion beam, which showed Ba/Ti ratio of 0.86 (± 0.04) (Fig 1 in the manuscript). We mentioned it as depth profiled XPS, but now clarified it in detail.

Now, we also show the XPS data from the surface (without etching, Fig R17a and b). The Ba/Ti ratio on the surface (i.e., w/o etch) is 0.68 (± 0.04), and it is indeed true that annealing and exposure to atmosphere changes the surface composition. We also observe the formation of BaCO₃ [1] on the surface which is evident from the high resolution XPS spectra of Ba 3d, O 1s and C 1s spectra as shown below. We now add this data to the SI (Fig S1d-f).

Fig R17 High resolution XPS data of the unetched BTO surface post annealing (a) Ba 3d (b) O1s (c) C 1s

We also performed STEM EDS spectrum imaging and quantified Ba/Ti ratio (Brown-Powell method) at various depths into the BTO film from the surface. As can be seen from Fig R18a and b, Ba/Ti ratio uniform throughout the film (if we ignore the first 2 nm at the surface). Ba/Ti~0.86 ± 0.04 (precision error), which matches well with the values determined by XPS (Fig R1b). The cross-sectional maps of various elements including the interfacial layer are presented in Fig R1c. Now we add this EDS quantification data to Fig S1a in SI. Using both XPS and EDS we confirm that our films are homogenous in composition and significantly Ba deficient.

Fig R18 (a) Cross section HAADF STEM of the BTO/TiN/Si stack after annealing in O₂ ambient showing different areas R1, R2 and R3, corresponding EDS atom% Ba/ Ti ratio from the regions marked in (a) is shown in (b), (c) shows the EDS mapping corresponding to image (a)

3. The authors state that the defective BTO reported in this manuscript is non-ferroelectric. The basis for this claim is the absence of switching in the AC I-V curves. However, Figure S5b shows some “leaky” ferroelectric-like switching characteristics. Furthermore, the authors only probe $\pm 6V$; is it possible the coercive voltage is above the $\pm 6V$ probing bias? Including additional characterization to verify the non-ferroelectric claims is vital (e.g. Polarization-electric field measurements/PUND, piezoresponse force microscopy, etc.).

Response: In Fig S5b (now Fig S12 in revised supplementary), we show the DC I-V characteristics of very leaky device. At best what it shows is memristive hysteresis, and not polarization switching. The maximum voltage we applied on devices is 8V, beyond which electrodes started peeling off. Our electrostrain response also saturates beyond 6 V, and thus we did not see any point in testing at larger voltages. See Fig R12 and response to referee 2’s corresponding question also.

We confirm that up until 8 V there is no ferroelectric switching (P-E obtained from instantaneous IV measurements), on all our devices. However, below 8 V we already see a second order electrostrain effect, which is a clear proof that ferroelectricity is not a reason for our second order electrostrain, and electrostriction is. We remain agnostic about the possibility of coercive field being above 8 V. We don’t measure beyond 8 V in order to not burn our devices.

However, just to be completely accurate, instead of saying that “our devices are not ferroelectric”, now we say in the manuscript that up until 8 V, our devices “did not show any ferroelectric switching”.

Minor comments:

1. Line 65: keep Gd:CeO₂ labels consistent (line 65 vs 58).

2. Figure 3: some axes labels overlap tick labels.

Thank you for pointing these errors. We now corrected them.

References

[1]. C.Miot et al, Residual Carbon Evolution in BaTiO₃ Ceramics Studied by XPS after Ion Etching, Journal of European ceramic society, 18, 1998, 339-343.

REVIEWER COMMENTS

Reviewer #1 (Remarks to the Author):

The authors made a lot of work for the revised manuscript. There is now doubt that the authors observe some electromechanical effect, which is not surprising for BaTiO₃. However, there is not nearly enough proof that the larger fraction of the effect is electrostriction.

1. To alleviate the doubts, it was suggested to measure the converse effect. That is to present a graph dielectric constant as a function of applied homogeneous stress. Moreover, it was suggested how to do it in a simple manner by bending of the substrate. Instead, the authors performed a very complicated experiment with nanoindentation, which does not provide a homogeneous stress and cannot be, therefore, meaningfully interpreted. The value of dielectric permittivity is not calculated, instead a tiny change in capacitance under load is presented in Fig R6. It is not informative. The value of dielectric permittivity is not shown anywhere.

2. The fact that the films show large strain hysteresis is inconsistent with electrostriction, but it is consistent with the tetragonal domain walls movement in BaTiO₃. True electrostrictors do not have permanent polarization and do not show mechanical hysteresis. This brings to a simple conclusion that the films are either ferroelectric or driven to the polar phase by the field. Moreover, the fact that the data presented clearly shows presence of the piezoelectric effect, makes the claim that the films are “non-ferroelectric” as stated in the title clearly erroneous. The fact that the response is dominant at the second harmonic with the absence of other data is not enough because of the strong hysteresis. Absence of a plot dielectric permittivity vs field adds to the fact that the data are not interpreted correctly.

3. Using the data provided, it is possible to estimate the change in curvature that causes 1 MPa stress, which according to the values of M₃₁ given, should lead to a change in the dielectric permittivity of ~100. The answer is 1/400 m. This is so little that it was enough to press on the substrate with a finger to cause measurable changes.

Given the data provided in the manuscript, in the supplementary and in the response to the reviewers, it is far more probable that the strain is the result of a very well-known effect: field induced cubic to tetragonal phase transition. At least this hypothesis explains all the data presented.

Reviewer #2 (Remarks to the Author):

The authors provide convincing responses to my comments and I have no further comments regarding the paper. I congratulate the authors on their work and the extensive response brought to the referees comments.

Reviewer #3 (Remarks to the Author):

The authors did a great job in revising the manuscript and addressing the points raised by the reviewers. I therefore recommend this work for publication.

We are glad that our extensive response and work that followed the first version of the manuscript indeed convinced referee 2 and 3 of our data and interpretations on giant electrostriction (M_{31} coefficients) in defective BaTiO₃. We thank them for their support to publish this manuscript as is.

We further thank referee 1 for more constructive feedback, and again very good questions. We performed the bending experiments that were suggested. We are happy to report that even these measurements substantiate our earlier claims of giant M_{31} coefficients measured. Furthermore, in the following, we will point out to data that rules out the alternate hypotheses proposed by referee.

The authors made a lot of work for the revised manuscript. There is now doubt that the authors observe some electromechanical effect, which is not surprising for BaTiO₃. However, there is not nearly enough proof that the larger fraction of the effect is electrostriction.

We hope that this round of review will also convince referee 1 of the validity of our claims.

1. To alleviate the doubts, it was suggested to measure the converse effect. That is to present a graph dielectric constant as a function of applied homogeneous stress. Moreover, it was suggested how to do it in a simple manner by bending of the substrate. Instead, the authors performed a very complicated experiment with nanoindentation, which does not provide a homogeneous stress and cannot be, therefore, meaningfully interpreted. The value of dielectric permittivity is not calculated, instead a tiny change in capacitance under load is presented in Fig R6. It is not informative. The value of dielectric permittivity is not shown anywhere.

On the vertical MIM stack device configuration, as the referee suggested bending experiments can be performed to estimate M_{31} , by estimating the change in dielectric constant of the active layer (ϵ_{33}) with applied stress (X_{11}), through a converse effect. The important point here is that the dielectric constant (or capacitance) of the active layer (BaTiO₃) changes significantly with the application of stress. However, it must be noted that this active layer (static $\epsilon_r=1500$ as per our *impedance fits shown in FigR1a and b*) is in series with a low dielectric constant ($\epsilon_r=15$) TiO_x, a passive layer, which dominates the total capacitance of the stack. As we explain, this will lead to only small changes in the total capacitance, be it in an indentation experiment or a bending experiment (although we did perform the experiment and report changes, see response to next questions).

To elucidate this, we show some back-of-the-notebook calculations of changes in stack capacitance to be expected (despite large changes in the capacitance of active layer) in a bending experiment with application of stress (100s of MPa) on a 120 nm thick film, in the following:

Given the device dimensions ($A=100 \times 100 \mu\text{m}^2$, $d_{\text{BaTiO}_3}=120 \text{ nm}$, $d_{\text{TiO}_x}=20 \text{ nm}$), and material dielectric constants (static $\epsilon_{33}\text{-BTO} = 1500$, static $\epsilon_{33}\text{-TiO}_x = 15$, see Fig R1), $C_{\text{TiO}_x} = 66.37 \text{ pF}$, $C_{\text{BTO}} = 1100 \text{ pF}$. The stack capacitance will then be ($1/C=1/C_{\text{BTO}}+1/C_{\text{TiO}_x}$) 62.59 pF , predominantly dominated by the low capacitance (high impedance) TiO_x layer. For this exercise, we ignore parallel resistors to these capacitors. If M_{31} (real part) of BTO is $10^{-15} \text{ m}^2/\text{V}^2$ (as reported), then at a stress of 650 MPa, change in dielectric constant of BTO is in the order

of ~ 73000 . Although, C_{BTO} (at 650 MPa) sees a huge change, the stack capacitance then is 65.92 pF, hardly a change by 3 pF from the unstressed conditions, again dominated by the low impedance TiO_x layer. The maximum stack capacitance is limited to 66.37 pF. Thus, we are looking at measuring changes in $\sim 1\text{-}3$ pF in the stack capacitance, irrespective of however huge the change in BTO capacitance/dielectric constant is (as we observed earlier in the nanoindentation measurements). As we reported earlier in the m/s, the sensitivity of our capacitance measurement is ~ 1 pF. As the referee suggested, we performed the bending experiments, with a maximum stress of 650 MPa on the film. In this experiment, assuming that the only reason for capacitance change in BTO is due to its M_{31} , we estimate that the minimum M_{31} we can measure is $\sim 10^{-16} \text{ m}^2/\text{V}^2$ (Fig R1c). Indeed, our bending experiments show a change in stack capacitance by ~ 2 pF, which corresponds to real part of M_{31} of at least $10^{-16} \text{ m}^2/\text{V}^2$ (also giant). The results are elucidated and discussed in detail as a response of subsequent questions.

Fig R1 (a) Bode impedance plot for vertical MIM device and corresponding fit using equivalent circuit in (b), (b) shows the values of RC elements obtained after impedance fitting (c) shows how increase in M coefficients saturates the stack capacitance that can be practically measured

The static dielectric constants of BTO and TiO_x layers were estimated from impedance fits, and now reported in the manuscript.

3. Using the data provided, it is possible to estimate the change in curvature that causes 1 MPa stress, which according to the values of M_{31} given, should lead to a change in the dielectric permittivity of ~ 100 . The answer is 1/400 m. This is so little that it was enough to press on the substrate with a finger to cause measurable changes.

To continue the discussion from question 1, we first respond to question 3. Thanks to the referee, we indeed performed bending experiments this time, but with the added caveat that despite large electrostrictive coefficients of the active layer, the changes in stack capacitance will be tiny. The domination of passive layer (TiO_x) in capacitance measurements was not considered by the referee in the estimations presented.

Fig. R2a shows the experimental set up with a fixture to bend the thin film stack on Si, while simultaneously performing impedance measurements. Initially, the sample (width=5 mm) is exactly fit in the fixture. The fixture is subsequently tightened using a screw, which reduces the distance between the two flat ends and thus bends the sample as shown in schematic in Fig. R2b. Motion of the screw by a pitch (360°) corresponds to reduction in the fixture dimension by 500 μm . Thus, the curvature of the film is controlled by the angle by which screw is tightened. We perform our experiments by tightening the screw from 0° to 5° , which corresponds to maximum radius of curvature of $1/(32.5 \text{ m})$. Assuming elastic modulus of BTO = 80 GPa, this corresponds to a max stress of $\sim 650 \text{ MPa}$.

Fig R2 (a) Experimental setup showing a fixture set up used for bending our sample and simultaneously measuring impedance using two probes, (b) schematic showing how the load is applied using the setup in (a), (c) capacitance as a function of frequency under 'no loading'(no bending) and 'loading state'(L1).

In the frequency range of 1 to 10 kHz in Fig R2c, we see that change in stack capacitance is $\sim 1\text{-}2 \text{ pF}$. At 650 MPa of stress, this corresponds to M_{31} in the order $10^{-16} \text{ m}^2/\text{V}^2$. This is also

giant (much more than classical response of BTO), but an order of magnitude smaller than our direct electrostrain measurements. Also, as we will expand in the next question, these bending experiments only give thermodynamic or real part of M_{31} . In the back-of-the notebook calculation shown in question 1, we assume ideal capacitors, no leakage, and unclamped conditions. The differences ($\sim 10^{-16}$ to 10^{-15} m²/V²) in the indirect and direct measurements can be because of all these factors. We now add this discussion following the indentation measurements in the supplementary information (Fig S7).

2. The fact that the films show large strain hysteresis is inconsistent with electrostriction, but it is consistent with the tetragonal domain walls movement in BaTiO₃. True electrostrictors do not have permanent polarization and do not show mechanical hysteresis. This brings to a simple conclusion that the films are either ferroelectric or driven to the polar phase by the field. Moreover, the fact that the data presented clearly shows presence of the piezoelectric effect, makes the claim that the films are “non-ferroelectric” as stated in the title clearly erroneous. The fact that the response is dominant at the second harmonic with the absence of other data is not enough because of the strong hysteresis. Absence of a plot dielectric permittivity vs field adds to the fact that the data are not interpreted correctly.

a) **“True electrostrictors do not have a permanent polarization”**

We disagree with this statement. All materials are electrostrictors, irrespective of whether they are polar or not. Effects of electrostriction are discussed even in polar materials such as GaN¹. Furthermore, what we are discussing here are non-classical electrostrictors.

b) **“Electrostrictors do not show hysteresis”**

Any hysteresis is indicative of underlying losses and dissipation. In this case, the losses refer to electromechanical losses, and can be estimated as $\tan \delta$ losses by following the phase of strain response in relation to E^2 (at 2ω). If strain follows E^2 in phase, then electrostrain curves show parabolae, otherwise they evolve into butterflies as shown in our illustration (now added in supplementary information movie, snapshots of this movie are shown in Figure R3a-c). We do not debate that our material is not lossy.

We do agree with the referee that electrostriction is a thermodynamic property (a third derivative of free energy), just like susceptibility or dielectric constant is (a second derivative). Thermodynamic properties are indeed lossless and conservative. They determine how much energy a material stores rather than dissipates. Despite that, in case of dielectric constant as a material property, it is difficult to find ideal dielectrics, and dielectric losses always exist. To include these losses in real world materials, we define a complex dielectric constant $\epsilon_{complex} = \epsilon_{real} + i\epsilon_{imag}$, where the real part (ϵ_{real}) describes the equilibrium thermodynamic property, and the imaginary part (ϵ_{imag}) describes the dissipative (non-equilibrium, entropy creating). Similarly, even in this work we estimate the complex M, in which the real part is lossless (hysteresis less), whereas the imaginary part of M describes the dissipation. **The thermodynamic part is only the real part.**

We represent $M_{31} = |M_{31}|e^{-i\delta}$ in the phasor notation, and quantified $|M_{31}|$ and $\tan\delta$ (Fig 3b and 3d in m/s). This can also be written as $M = M_{real} + i M_{imaginary}$. Now we show the values of $M_{31-real}$ as a function of frequency (in Fig 3b with $|M_{31}|$, also see in SI Fig S12c and d). These

are also in the order of $10^{-15} \text{ m}^2/\text{V}^2$. Infact, this component of the complex M corresponds to lossless, hysteresis less electrostriction. Furthermore, the converse thermodynamic measurements of measuring electrostriction by measuring the change in susceptibility by applying stress corresponds to only $M_{31\text{-real}}$ (Fig R3d-f).

In our devices, we show that at low frequencies ($<1000 \text{ Hz}$), strain follows E^2 almost in phase, with δ being quite small. In other words, $M_{31\text{-real}}$ dominates the electrostrictive response. As frequency increases (especially between 5-8 kHz for various devices), losses ($\tan\delta$) increase and peak (Fig R3d inset). Even there we show that while the magnitude of reported $|M_{31}|$ is larger than $M_{31\text{-real}}$, they are still in the same order of magnitude ($10^{-15} \text{ m}^2/\text{V}^2$). ***In other words, the equilibrium lossless electrostrictive component of complex electrostriction tensor element, itself is giant.***

The relaxation in electrostriction is shown to be correlated with the dielectric relaxation, including in the loss. Further the existence of NPR-like regions (Fig R3g, also Fig 1c), and some 2D electroactive defects (Fig R3h, also Fig S2c), suggest that the electroactive nature of these defects contributes to the observed non-classical electrostriction. We propose that the fundamental origin of all these relaxations and the correlated giant electrostrictive behavior at these frequencies is due to structural and polarization disorder created by Ba and oxygen non-stoichiometry, and associated lattice anharmonicity. The giant $|M_{31}|$ coefficients at larger frequencies are a consequence of large electroactive defect induced polarizabilities, coupled elastic dipoles and long-range coherent strain fields in addition to possible electroactive twin wall motion (just like ferroelectric domain boundary motion in ferroelectrics, except that our samples are not ferroelectric).

c) The fact that the response is dominant at the second harmonic with the absence of other data is not enough because of the strong hysteresis.

Hysteresis does not affect the strength of second harmonic response. Hysteresis is observed in field vs strain plots, where the strength of second harmonic response is obtained from strain vs time plots. Hysteresis (and butterfly loops) arises as a result of phase lag between the response (in this case displacement/strain) and the cause (E^2 , whose AC component has a 2ω frequency, see the supplementary video now, snapshots in Fig R3a-c). From the raw signal, we filter out the second harmonic response and then understand the hysteresis only at 2ω frequency. Also, as discussed previously, hysteresis is a result of the imaginary part of M. So, here we show in Fig R3(d-f) hysteresis free “giant” $M_{31\text{-real}}$ as a function of frequency on three different devices of various leakages.

Furthermore, as referee 2 pointed out in our previous round of review, a piezoelectric effect (with losses and hysteresis) shows up in the frequency spectrum as all the odd harmonics (not just first). Electrostrictive response on the other hand shows up as all the even harmonics. We clearly show not only a second harmonic response but also a 4th harmonic response (Fig S5b, also shown here as Fig R3i), very weak first order response (which is also not consistent), and zero third order response. Our data is strong enough to claim that our system is electrostrictive.

d) Absence of a plot dielectric permittivity vs field adds to the fact that the data are not interpreted correctly.

This data was also presented as capacitance vs voltage at different frequencies in Fig S9f in the lateral devices. Capacitance decreases with field symmetrically, reminiscent of relaxor-like

materials. Furthermore, no butterfly hysteresis is observed here, further strengthening the absence of ferroelectricity in our samples.

Fig R3 Snapshots (a) (b) and (c) at three different phase differences and the evolution of hysteresis in second order butterfly plot (red curve represents voltage waveform, purple curve is second order displacement waveform and blue curve is displacement time plot). $|M_{31}|$ and $M_{31-real}$ shown as a function of frequency for three different devices in (d) (e) and (f). (g) High resolution HAADF STEM image of BTO overlaid with polarization map where nano polar-like regions (NPRs) are enclosed in white boxes (h) 2D electroactive defects present as twin boundaries (i) fourier transform of displacement response driven at 1 kHz showing strong presence of second harmonics and its multiple harmonics while the first order response is weak even smaller than fourth harmonic.

Given the data provided in the manuscript, in the supplementary and in the response to the reviewers, it is far more probable that the strain is the result of a very well-known effect: field induced cubic to tetragonal phase transition. At least this hypothesis explains all the data presented.

Application of field does orient nanopolar domains in the direction of the field. But this is not a phase transition. A phase transition should be accompanied by anomaly in capacitance vs field measurements, which we do not see (Fig R4a, also in Fig S9f). Furthermore, our polarization vs field measurements, and corresponding instantaneous current vs field measurements (presented in Fig 4a, also shown here as Fig R4b) do not show any peculiar

peaks at a critical field for the proposed phase transition. Since we do not see any indications, our data clearly does not support this hypothesis.

Fig R4 (a) Capacitance as a function of voltage at various frequencies showing no anomaly in capacitance (b) I-V characteristic showing no peculiar peak.

Our data further rules out Joule heating and ferroelectric switching, the other possible second order electrostrain effects convincingly. As a result, according to us (and referee 2 and 3), our proofs only point towards giant electrostriction.

References

1. Muensit, S. & Guy, I. L. Electromechanical effects in gallium nitride. *Ferroelectrics* **262**, 195–200 (2001).

Reviewer #2 (Remarks to the Author):

The authors provide convincing responses to my comments and I have no further comments regarding the paper. I congratulate the authors on their work and the extensive response brought to the referees comments.

Reviewer #3 (Remarks to the Author):

The authors did a great job in revising the manuscript and addressing the points raised by the reviewers. I therefore recommend this work for publication.

We again thank reviewers 2 and 3 for their comments and earlier feedback to make this a real high quality work.

REVIEWER COMMENTS

Reviewer #1 (Remarks to the Author):

I am ready to agree with the authors on most of the issues, however, the main point remains unsolved:

NON-ferroelectric electrostrictors exhibit mechanical hysteresis ONLY in the vicinity of the relaxation frequency. For instance, for PMN-PT15, this frequency is about 6-8 kHz. For PMN-PT10 it is higher (up to 10-18 kHz but the strain is lower). Dielectric relaxation and mechanical relaxation occur close together. The authors can purchase a sample of PMN-PT with less than 15% PT and check it by themselves.

Absence of mechanical hysteresis at low frequencies is the most important practical advantage of non-ferroelectric electrostrictors.

Mechanical hysteresis in electrostrictors appears when there is some remanent polarization (i.e. ferroelectricity) leading to a domain wall motion. In this view Fig R2c in the rebuttal layer is incompatible with Fig. 3a in the text if the electrostriction effect is assumed: Fig R2c shows no relaxation whatsoever, while Fig. 3a shows large hysteresis. It is, however, fully compatible with the domain wall motion. It is much simpler explanation to all data presented.

Reviewer #4 (Remarks to the Author):

I will first comment on the last remarks of Referee no. 1 and then provide my own comments on the paper, which I believe are important.

The referee claims that the large hysteresis in the strain-voltage response indicates a strain mechanism that is not related to electrostriction but to some other common electro-mechanical effect, such as domain wall motion.

The issue here is, in fact, rather complex. First, the referee is incorrect in stating that electro-mechanical hysteresis does not appear at low frequencies in non-ferroelectric materials. The referee also refers in their argument to Fig. R2c when the reference should be given to Figure 4b in the text, which does show relaxation in capacitance. It is true that the hysteresis seen by the authors is large, but nonzero hysteresis is also observed in classical electrostrictors such as PMN-0.1PT in the strain-electric field relation. See, for example, Refs. 1 and 2. It is the relationship between strain and polarization squared, which in classical electrostrictors, does not exhibit hysteresis and nonlinearity. I will come back to this point later.

Second, under sub-switching conditions, the motion of ferroelastic-ferroelectric domain walls does not contribute to the electrostriction in ferroelectrics, only to the piezoelectric effect. The referee is therefore partly mistaken here as well. However, the motion of ferroelectric (180°) domain walls does contribute to the second harmonic, i.e., it would be a contribution to the electrostrictive response. But so does switching of polar-nano regions in relaxors, which is probably the origin of the small hysteresis in PMN-based electrostrictors. The authors claim that they do not see domain switching in their experiments. The effect then seems to be dominated by motion (switching?) of polar nano region (nano domains?), which authors have evidenced and lattice deformation associated with the presence and motion of ionic vacancies. The first mechanism would be similar to that in PMN-based electrostrictors while the second one could be similar to that recently reported in oxides with fluorite structure.

The author's message that, I think, the referee does not see is that the authors describe what should be non-classical electrostriction. In these so-called "giant" electrostrictors, the mechanism of electrostriction is augmented by the electro-mechanical response of ionic vacancies (Ba and O in this case and maybe Ti^{+3} polarons) rather than the perfect lattice. One can argue that electrostriction in relaxors is also dominated by defects, which in that case are polar nano regions. This is true, but the way in which polar nano regions distort the lattice is different than vacancies do. The contribution to strain from chemical expansion due to the vacancies and motion of the vacancies under an electric field is not yet well understood in oxides. Therefore, the explanations of the "giant" electrostriction are still hand-waving, and this paper is not different. In this sense, it may be advisable to call the "giant electrostriction effect" described in this and other papers a "giant electrostriction-like effect."

From this point of view, I therefore do not think that the referee's objections are completely valid.

However, I have a few arguments of my own that I think are fundamental and that have not been addressed properly or at all in the previous rounds of the reviews.

The first is minor and along the lines drawn by Referee 1 and concerns the fact that the strain-polarization relationship is strongly hysteretic, which is unexpected. What it indicates is that there are contributions to the dielectric displacement D that are not mechanically active. The hysteresis in strain-polarization disappears when all mechanisms to electrostriction that contribute to strain also contribute to polarization. The authors should also note that the area of electro-mechanical hysteresis (either first or second harmonic) does not have units of energy and therefore does not necessarily represent energy loss. But this is a minor point.

I now have the following more serious comments on the paper.

The first concerns the value of the measured Q coefficient, which is estimated by the authors to be 10^{-7} (C/m²)². This is about 5 orders of magnitude lower than what is known for bulk BaTiO₃. I do not see how 15% of defects can change this fundamental material constant by that much. The reason for this discrepancy is probably the wrong estimate of the field-induced polarization (or dielectric displacement, which is not the same). It is much easier for me to accept a giant M than five orders of magnitude reduced Q values.

The second concerns the value of the longitudinal M₃₃ coefficient, which has its normal value, that is about four orders of magnitude lower than M₃₁. Such a degree of anisotropy would be unusual for BaTiO₃ and suggests that some non-electro-mechanical effect has a role in the measured strain.

This leads us to the third argument I want to make. When looking at the measurement configuration, it is seen that electrical conductivity during M₃₃ measurements is much lower than for M₃₁ measurements (because of the TiO layer). This suggests that Joule heating may play a role in measurements of M₃₁ but not M₃₃. I see that the authors have discussed in detail thermal effects. The question I have is the following: when authors modeled thermal response, have they considered heating of the substrate from the heat coming from the film? One can easily calculate that if the substrate heats by only 0.5K, the 500 μm thick Silicon substrate would thermally expand by 500 pm. These are displacement values that authors measure in their experiments. I actually suspect that in most of the reports of the giant electromechanical response, thermal expansion has a non-negligible role.

Two more minor points: The authors write, “Given that the 3rd order response is also absent, we conclude that our material is not piezoelectric or very weakly piezoelectric, and thus in the rest of the discussion, we do not analyze piezoelectricity in any further detail,” and then, “To better compare with conventional piezoelectric materials also, we report the values of d₁₃-effective (d₁₃*) estimated as Max Strain/Max field as a function of frequency in Fig S5d. d₁₃* is 2.57 nm/V at 1 kHz and at higher frequency such as 9 kHz, it decreases to 390 pm/V. These values are larger or comparable with Pb-based materials that show larger electrostrain.” These two statements are contradictory.

Finally, the authors write, “The second-order EM behavior is an effect of one or more of the following phenomena: a) ferroelectric switching, b) thermal expansion because of device heating, c) non-classical defect-induced electrostriction.” What about intrinsic, lattice electrostriction – it is present in all materials?

References :

1. Cross, L. E.; Jang, S. J.; Newnham, R. E.; Nomura, S.; Uchino, K. Large Electrostrictive Effects in Relaxor Ferroelectrics. *Ferroelectrics* 1980, 23, 187–192.

2. Newnham, R. E.; Sundar, V.; Yimnirun, R.; Su, J.; Zhang, Q. M. Electrostriction: Nonlinear Electromechanical Coupling in Solid Dielectrics. *The Journal of Physical Chemistry B* 1997, 101 (48), 10141–10150. <https://doi.org/10.1021/jp971522c>.

We thank referee 4 for describing in detail (in better words than we could) to rebut the arguments of referee 1. We also thank the referee for raising other pertinent points. In the following, we provide a response for every question. We hope the referee and the editor are satisfied with this round of review.

We also are grateful and glad about the high quality of peer review by all the referees in all the three rounds of reviews. This has definitely made the paper much stronger and given us leads to follow in the future in this direction.

Response to referee #4

I will first comment on the last remarks of Referee no. 1 and then provide my own comments on the paper, which I believe are important.

The referee claims that the large hysteresis in the strain-voltage response indicates a strain mechanism that is not related to electrostriction but to some other common electro-mechanical effect, such as domain wall motion.

The issue here is, in fact, rather complex. First, the referee is incorrect in stating that electro-mechanical hysteresis does not appear at low frequencies in non-ferroelectric materials. The referee also refers in their argument to Fig. R2c when the reference should be given to Figure 4b in the text, which does show relaxation in capacitance. It is true that the hysteresis seen by the authors is large, but nonzero hysteresis is also observed in classical electrostrictors such as PMN-0.1PT in the strain-electric field relation. See, for example, Refs. 1 and 2. It is the relationship between strain and polarization squared, which in classical electrostrictors, does not exhibit hysteresis and nonlinearity. I will come back to this point later.

Second, under sub-switching conditions, the motion of ferroelastic-ferroelectric domain walls does not contribute to the electrostriction in ferroelectrics, only to the piezoelectric effect. The referee is therefore partly mistaken here as well. However, the motion of ferroelectric (180°) domain walls does contribute to the second harmonic, i.e., it would be a contribution to the electrostrictive response. But so does switching of polar-nano regions in relaxors, which is probably the origin of the small hysteresis in PMN-based electrostrictors. The authors claim that they do not see domain switching in their experiments. The effect then seems to be dominated by motion (switching?) of polar nano region (nano domains?), which authors have evidenced and lattice deformation associated with the presence and motion of ionic vacancies. The first mechanism would be similar to that in PMN-based electrostrictors while the second one could be similar to that recently reported in oxides with fluorite structure.

The author's message that, I think, the referee does not see is that the authors describe what should be non-classical electrostriction. In these so-called "giant" electrostrictors, the mechanism of electrostriction is augmented by the electro-mechanical response of ionic vacancies (Ba and O in this case and maybe Ti³⁺ polarons) rather than the perfect lattice. One can argue that electrostriction in relaxors is also dominated by defects, which in that case are polar nano regions. This is true, but the way in which polar nano regions distort the lattice is different than vacancies do. The contribution to strain from chemical expansion due to the vacancies and motion of the vacancies under an electric field is not yet well understood in oxides. Therefore, the explanations of the "giant" electrostriction

are still hand-waving, and this paper is not different. In this sense, it may be advisable to call the “giant electrostriction effect” described in this and other papers a “giant electrostriction-like effect.”

We thank the current referee for these great insights. Indeed, we have the same position, but did not want to speculate the exact mechanism in the paper at this time. We just correlate structural features with the properties, but do not speculate the exact causation. We are currently designing more experiments (including in situ microscopy) to understand these mechanisms in detail and will report this in future. But this paper is about reporting that such a giant electrostrain effect exists in our samples and we want to send out the message that given these large effects, our system in particular, and probably non-stoichiometric perovskites in general are good materials platforms to study these effects.

We accept the current referee’s suggestion, and will call the effect, “giant electrostriction-like effect”. We have now modified the manuscript to reflect this.

From this point of view, I therefore do not think that the referee’s objections are completely valid.

However, I have a few arguments of my own that I think are fundamental and that have not been addressed properly or at all in the previous rounds of the reviews.

The first is minor and along the lines drawn by Referee 1 and concerns the fact that the strain-polarization relationship is strongly hysteretic, which is unexpected. What it indicates is that there are contributions to the dielectric displacement D that are not mechanically active. The hysteresis in strain-polarization disappears when all mechanisms to electrostriction that contribute to strain also contribute to polarization. The authors should also note that the area of electro-mechanical hysteresis (either first or second harmonic) does not have units of energy and therefore does not necessarily represent energy loss. But this is a minor point.

This point is well-taken and noted. We completely agree with the interpretation that mechanisms responsible for D are not mechanically active and thank the referee for this wonderful insight. We confess that, this completely escaped us.

Our films show giant M_{31} , (field-strain measurements and corresponding hysteresis, and the converse measurements), and this is what we want the readers to take as the main message. The strain-polarization curves were added as a response to referee 2’s comments in round 1. We agree that there is much more to understand in this data and will pursue it as further studies.

I now have the following more serious comments on the paper.

The first concerns the value of the measured Q coefficient, which is estimated by the authors to be $10^{-7} (C/m^2)^2$. This is about 5 orders of magnitude lower than what is known for bulk BaTiO₃. I do not see how 15% of defects can change this fundamental material constant by that much. The reason for this discrepancy is probably the wrong estimate of the field-induced polarization (or dielectric displacement, which is not the same). It is much easier for me to accept a giant M than five orders of magnitude reduced Q values.

Fig R1: Empirical relation between M and Q coefficient. Taken from reference¹

This is a very valid concern. It is indeed true that we can overestimate the values of dielectric displacement (D), given the leakage. This is the reason why we report the Q values only in less-leaky devices, and only sources upto 1 to 3 V, where leakage effects are very small. We also included the discussion on Q following referee 2's comments. We want to make it clear that the main message that the paper gives out is on M coefficients.

Having said that, we report an enhancement in M by 3-4 orders of magnitude depending on the frequency of operation, and reduction in Q by 5 orders of magnitude. Although, initially it was surprising, we convinced ourselves that this may be possible given the observations in the following papers^{1,2}:

1. Yu, J. & Janolin, P.E. *Defining 'Giant' Electrostriction*. *J Appl Phys* **131**, (2021).
2. Newnham, R. E., Sundar, V., Yimmirun, R., Su, J. & Zhang, Q. M. Electrostriction: Nonlinear Electromechanical Coupling in Solid Dielectrics, *J. Phys. Chem. B*, **101**, 10141-10150 (1997).

To quote from the Newnham paper (ref 1 above), " Q values vary in an opposite way to M values. Q ranges from $10^{-3} \text{ m}^4/\text{C}^2$ in relaxor ferroelectrics to greater than the order of $10^3 \text{ m}^4/\text{C}^2$ in polyurethane films", and this relates to the trend on the negative slope part of **Fig R1**. Our material behaves like relaxor ferroelectric, and so if one were to extrapolate the M values of relaxors in **Fig R1** (on the left side) to 10^{-14} to $10^{-15} \text{ m}^2/\text{V}^2$, we see that the lower limit of Q for classic electrostrictors will be 10^{-6} to $10^{-5} \text{ m}^4/\text{C}^2$. Since we have non-classical electrostrictor, it was OK for us that this trend is slightly deviated from. However, we do take the referees point about overestimating the values of D because of slight leakage, and now in the paper suggest that our reported values of Q are underestimated. In the future we will report on trends in Q , by taking care of this issue, but we emphasize that in this paper the main message is about M .

If the referee is still uncomfortable with these arguments, we do not mind removing the discussion on Q as was our first version before referee 2 asked us to include it. But we are of the opinion that this data will serve as prelude to invite discussion in future works.

The second concerns the value of the longitudinal M33 coefficient, which has its normal value, that is about four orders of magnitude lower than M31. Such a degree of anisotropy would be unusual for BaTiO₃ and suggests that some non-electro-mechanical effect has a role in the measured strain.

About anisotropy, we argue that unlike in bulk, anisotropy arises because the film is on a substrate. Another property, we measured for some other work on the same samples is coefficient of thermal expansion, which clearly shows anisotropy (CTE out of plane: 2.4×10^{-5} /K, CTE in-plane is 4×10^{-6} /K, published in ACS Electronic Materials³). Such an anisotropy is due to substrate clamping effects. Next, we are trying to release these devices to build MEMS based cantilevers, which will give us unconstrained coefficients of our defective BTO. In the following question, we rule out the effect of Joule heating on the substrate that the referee proposed as possible non-electromechanical effect producing the strain that we measure.

This leads us to the third argument I want to make. When looking at the measurement configuration, it is seen that electrical conductivity during M33 measurements is much lower than for M31 measurements (because of the TiO layer). This suggests that Joule heating may play a role in measurements of M31 but not M33. I see that the authors have discussed in detail thermal effects. The question I have is the following: when authors modeled thermal response, have they considered heating of the substrate from the heat coming from the film? One can easily calculate that if the substrate heats by only 0.5K, the 500 μ m thick Silicon substrate would thermally expand by 500 pm. These are displacement values that authors measure in their experiments. I actually suspect that in most of the reports of the giant electromechanical response, thermal expansion has a non-negligible role.

This is a good point. Here we present some experimental data, and simulations to show that the effect of substrate in the measured strain oscillations is negligible.

Ours is a dual beam measurement, with reference beam shone about a few micrometers away from the device. Thermal expansion of the substrate will not be restricted to only to the device area since heat transport occurs in longer range, and so a reference point close to the device should capture the substrate expansion and contraction with thermal cycling. However, our differential measurement should take care of it, and eliminate any substrate contribution.

Having said that, in *Fig R2a-e*, we show data on five different devices of variable leakage measured at different frequencies where we show displacement measured from reference laser, the measurement laser, and the difference signal (not differential but difference). We clearly see that the contribution from the reference laser (to be understood as substrate contribution) is negligible in all the cases.

Fig R2: Raw displacements and reference measurement for excluding the thermal expansion effects shown for the “less-leaky” device at 3kHz (in (a)), 5kHz (in (b)), and “more-leaky” device at 1kHz, 3kHz and 5kHz (in c-e) respectively.

To exactly understand the temperature changes on the region of the substrate directly under the device, we also performed electrothermal simulations. In these simulations (**Fig R3**), to model the heat transport in the vertical direction of Si, we consider Si as

- (i) Only a single thermal resistor of thickness 500 μm in one case (model shown in **Fig R3a** and results shown in **c(left)** and **d(left)**, this does not account for various depths of the substrate being at different temperatures) and
- (ii) split it as two thermal resistors in series of thickness 250 μm each (model shown in **Fig R3b** and results shown in **c(right)** and **d(right)**). This strategy is standard in LT Spice modelling i.e. more the sections, more precisely can the temperature at various depths of Si be modelled.

We can see that in our less-leaky representative devices, change in temperature of Si is of the order of a mK, corresponding to a displacement of 2.5 μm in amplitude at 1 kHz, and in case of leakier devices it is <10 mK, corresponding to a displacement of <20 μm at 1 kHz. The substrate temperature rise (at different points in the substrate), will be even smaller if more sections are included in the simulations. This is noise compared to the measured amplitudes of the device. In general, it is reasonable and well-known that Si (substrate) is a very good heat reservoir.

These reasons show that substrate thermal expansion plays a negligible role in the effects that we have shown. We now add this discussion and data to the SI (Fig S16 and S17 and corresponding supplementary note 14).

Fig R3: Electrothermal simulations for estimating the temperature rise in Silicon substrate for (a) electrothermal circuit with single 500 μm Si layer (b) with double Si layer 250 μm each (additional layer in dotted highlight) (c) simulated temperature rise for less leaky device in single (left) and double Si layer (right) (d) temperature rise for leakier device in single (left) and double Si layer (right).

Two more minor points: The authors write, “Given that the 3rd order response is also absent, we conclude that our material is not piezoelectric or very weakly piezoelectric,

and thus in the rest of the discussion, we do not analyze piezoelectricity in any further detail,” and then, “To better compare with conventional piezoelectric materials also, we report the values of d13-effective (d13*) estimated as Max Strain/Max field as a function of frequency in Fig S5d. d13* is 2.57 nm/V at 1 kHz and at higher frequency such as 9 kHz, it decreases to 390 pm/V. These values are larger or comparable with Pb-based materials that show larger electrostrain.” These two statements are contradictory.

The $d_{13_effective}^*$ data was added as a response to referee 2’s comments that suggested that piezoelectric responses are better understood by readers rather than electrostrictive responses. So to compare our electrostrain values at given fields, if one can define an effective piezo response as Max Strain/Max voltage, it gives a like to like metric to compare with standard piezoelectric materials. However, it is not our intention to say that our material is piezoelectric or shows linear response. We now clarified this in the manuscript.

Finally, the authors write, “The second-order EM behavior is an effect of one or more of the following phenomena: a) ferroelectric switching, b) thermal expansion because of device heating, c) non-classical defect-induced electrostriction.” What about intrinsic, lattice electrostriction – it is present in all materials?

Thank you for pointing this out. Now we also add field induced phase transitions (of which ferroelectricity is a subset) and intrinsic lattice electrostriction as two more effects that will comprehensively describe all possible second order effects.

References

1. Yu, J. & Janolin, P.E. Defining ‘Giant’ Electrostriction. *J Appl Phys* **131**, (2021).
2. Newnham, R. E., Sundar, V., Yimnirun, R., Su, J. & Zhang, Q. M. Electrostriction: Nonlinear Electromechanical Coupling in Solid Dielectrics, *J. Phys. Chem. B* , **101**, 10141-10150 (1997).
3. Vura, S. *et al.* Epitaxial BaTiO₃ on Si(100) with In-Plane and Out-of-Plane Polarization Using a Single TiN Transition Layer. *ACS Appl Electron Mater* **3**, 687–695 (2021).

REVIEWERS' COMMENTS

Reviewer #4 (Remarks to the Author):

The authors have addressed all the points I raised and provided satisfactory arguments. The paper presents important results that should inspire further fundamental and experimental studies in the field of giant electro-mechanical response